# CLASS-WISE AUTOENCODERS MEASURE CLASSIFICATION DIFFICULTY AND DETECT LABEL MISTAKES

## ABSTRACT

We introduce a new framework for analyzing classification datasets based on the ratios of reconstruction errors between autoencoders trained on individual classes. This analysis framework enables efficient characterization of datasets on the sample, class, and entire dataset levels. We define reconstruction error ratios (RERs) that probe classification difficulty and allow its decomposition into (1) finite sample size and (2) Bayes error and decision-boundary complexity. Through systematic study across 19 popular visual datasets, we find that our RER-based dataset difficulty probe strongly correlates with error rate for state-of-the-art (SOTA) classification models. By interpreting sample-level classification difficulty as a label mistakenness score, we further find that RERs achieve SOTA performance on mislabel detection tasks on hard datasets under symmetric and asymmetric label noise.

## 1 INTRODUCTION

Data is the cornerstone of modern machine learning. As the data-centric AI movement has made increasingly clear, both predictive and generative ML models rely on sufficiently large and diverse high-quality datasets (Deng et al., 2009b; Radford et al., 2018; Kaplan et al., 2020). However, it is well known that even popular visual datasets like CIFAR-100 (Krizhevsky & Hinton, 2009), Caltech-256 (Griffin et al., 2007), and ImageNet (Deng et al., 2009b) can have hundreds or thousands of data quality issues, including up to $10\%$ label errors (Northcutt et al., 2021). Consequently, curating a high-quality dataset requires not only data collection but also data cleaning, characterization, evaluation, and refinement.

Nevertheless, existing methods for data quality assessment are inherently limited. Methods that seek to estimate the classification difficulty of a sample or dataset are either model-dependent (Ethayarajh et al., 2021), computationally infeasible (Scheidegger et al., 2021), or break down when applied to challenging datasets (Zhang et al., 2020). Likewise, mislabel detection methods either rely on training a strong classifier on the dataset (Pruthi et al., 2020; Pleiss et al., 2020), which becomes more time and compute-intensive for more complex datasets, or exhibit degraded performance on datasets with complex decision boundaries (Zhu et al., 2021; Northcutt et al., 2021).

To address these limitations, we propose a novel approach for characterizing the difficulty of classification datasets by decomposing complex multi-class classification problems into one manifold learning problem for each class. Explicitly, we generate a feature vector for each sample from a foundation model like CLIP ViT-B/32 (Radford et al., 2021), train a shallow autoencoder on the feature vectors for each class. We call these autoencoders reconstructors, as they are used to capture how well a new sample is reconstructed by the shallow model. We then compute the reconstruction error for each sample with respect to each reconstructor, and use ratios of these reconstruction errors to estimate the difficulty of individual samples, classes, subsets, and entire datasets.

This method, which we call Reconstruction Error Ratios (RERs), is theoretically motivated, intuitive, and offers several key advantages:

*Efficiency*: Reconstructors can be trained in seconds, and training and inference can be parallelized over CPU cores. Further acceleration can be achieved with minimal reduction in performance by fitting the reconstructors on a fraction of the data — in many cases we observe SOTA performance when fitting on just 100 samples per class.

*Interpretability*: RERs allow us to compare the relative difficulty of specific samples, entire classes, data subsets, and entire datasets. They enable dataset-wide error rate estimation, and provide principled label mistake probabilities for each sample.

*Generality*: RERs provide a unified pipeline for processing datasets of different sizes and modalities, and work with features from any foundation model. They also extend readily to challenging datasets and datasets with arbitrarily many classes.

RERs perform remarkably well in both classification difficulty and mislabel detection tasks. Through a comprehensive study across 19 visual datasets, we demonstrate strong correlations between RER-based difficulty measures and state-of-the-art classification error rates. By interpreting sample difficulty scores as mislabel likelihood scores and employing a simple threshold ansatz to classify samples as mistaken, we find that RERs outperform other feature-based mislabel detection techniques under various noise conditions.

Our primary contributions are as follows:

1. A formal framework for applying Reconstruction Error Ratios for dataset analysis.
2. Empirical validation of RERs as a measure of the difficulty of classification.
3. A method for decomposing classification difficulty into distinct components representing finite-size contributions and Bayes error and decision-boundary contributions.
4. Demonstration of RERs' efficacy in mislabel detection tasks.

We believe that this work is a significant step forward in the direction of principled dataset analysis.

## 2 BACKGROUND AND RELATED WORK

Our work intersects with several areas of machine learning research, including dataset difficulty assessment, autoencoder applications, and mislabel detection. In this section, we review relevant literature in these domains and contextualize our contributions.

### 2.1 DATASET DIFFICULTY

Understanding and quantifying the difficulty of classification tasks has long been a challenge in machine learning. Early work in the visual domain by (Ionescu et al., 2016) focused on human response times as a measure of image classification difficulty. While informative, this approach is not scalable and does not address dataset-level challenges.

(Ho & Basu, 2002) propose using geometric properties of datasets to assess difficulty, but focused primarily on binary classification tasks in low-dimensional feature spaces. Through a UMAP graph-layout loss term, our method also utilizes geometric information to estimate dataset difficulty, and generalizes well to classification problems with many classes in high-dimensional feature spaces.

More recently, information-theoretic approaches like DIME (Zhang et al., 2020) and $\mathcal{V}$-Usable Information (Ethayarajh et al., 2021) have shown promise. However, the former gives only upper bounds, ruling out strict ordering, and the latter is model-dependent, limiting its generalizability. Finally, (Scheidegger et al., 2021) explore using silhouette scores and FID scores for dataset difficulty assessment and introduce shallow classifiers called probe nets whose error correlate strongly with larger classification models. Our RERs are defined similarly to their silhouette score-based difficulty scores, offer faster computation than any of these methods, are more interpretable, and correlate as if not more strongly with error rate of state-of-the-art models.

### 2.2 AUTOENCODERS AND THEIR APPLICATIONS

Autoencoders have a rich history in machine learning, dating back to the work of (Rumelhart et al., 1986; Bourlard & Kamp, 1988; Hinton & Zemel, 1993). They have been used for dimensionality reduction, feature learning, and generative modeling. Variants such as denoising autoencoders (Vincent et al., 2008) and variational autoencoders (VAEs) (Kingma & Welling, 2022) have further expanded their capabilities, and they are even used in the pretraining of diffusion models (Rombach et al., 2021).

Autoencoders have also been used in the context of visual anomaly detection, where autoencoders trained on normal data can identify anomalous samples by their high reconstruction errors. Our work differs by using class-wise autoencoders to assess intra-class and inter-class similarities, focusing on classification difficulty rather than anomaly detection. Furthermore, we perform autoencoding on the features from a foundation model like CLIP (Radford et al., 2021) and DINOv2 (Oquab et al., 2024), rather than on images themselves.

### 2.3 MISLABEL DETECTION

Mislabel detection seeks to identify erroneous labels in a dataset, with approaches falling into two main categories: (1) feature-based approaches like SimiFeat (Zhu et al., 2021) and (2) training-based approaches like (Pleiss et al., 2020) and TracIn (Pruthi et al., 2020), which are time-intensive and require access to the training dynamics.

Confident Learning (Northcutt et al., 2021) is a popular approach that uses any classifier trained on a given dataset to estimate the joint distribution of noisy and true labels. A feature-oriented variant of Confident Learning was recently found to achieve comparable performance when training a simple logistic regression classifier on CLIP features (Srikanth et al., 2023b).

Like (Zhu et al., 2021) and (Srikanth et al., 2023b), our RER-based approach is feature-based, but it differs from these methods by decomposing high-dimensional classification tasks into low-dimensional class-specific manifold learning problems, offering an efficient alternative that achieves better performance on hard datasets.

## 3 THE RECONSTRUCTION ERROR RATIO

In this work, we focus our attention on supervised classification settings. In this context, reconstruction errors and their ratios are defined with respect to a dataset consisting of features and labels

$$D = (\boldsymbol{X}, \boldsymbol{y}), \tag{1}$$

where $\boldsymbol{X} \in \mathbb{R}^{N \times d}$ is a matrix of $d$-dimensional features for each sample, $\boldsymbol{y} \in \{0, 1, \ldots, N_c - 1\}^N$ is a vector containing a single integer-valued label for each sample, $N$ is the number of samples, and $N_c$ is the number of classes.

Whereas typical image classification problems treat a preprocessed and flattened version of the image to be classified as the input features, we instead use $\boldsymbol{X}_{j,:}$ to denote the feature vector obtained by feeding image $j$ through a visual foundation model like CLIP ViT-B/32 or DINOv2-B. This allows for unified processing and comparison across datasets.

A sample from the dataset is a feature-label pair, $D_j = (\boldsymbol{X}_{j,:}, \boldsymbol{y}_j)$. We assume that $(\boldsymbol{X}_{j,:}, \boldsymbol{y}_j)$ are random variables drawn from distribution $(\mathcal{X}, \mathcal{Y})$. These labels may contain noise, either in the form of ambiguity or swapped labels. When indices are not needed, we use the streamlined notation $(\boldsymbol{x}, y)$ to refer to a general feature-label pair.

Our high-level goal is to characterize the dataset $D$ without training a (potentially large) classification model on $D$. Towards that end, we decompose the dataset by class and use shallow autoencoders to learn robust representations of these class manifolds.

Let $X^c = \{\boldsymbol{x} = \boldsymbol{X}_{j,:} | \boldsymbol{y}_j = c\}$ denote the subset of features in the dataset that have assigned (potentially noisy) label $c$. For each class, we train an encoder-decoder pair $(f, g)$, where $f : \mathbb{R}^d \to \mathbb{R}^{d_{latent}}$ and $g : \mathbb{R}^{d_{latent}} \to \mathbb{R}^d$, such that

$$r(\boldsymbol{x}) = g(f(\boldsymbol{x})), \tag{2}$$

is the reconstruction function. Each class autoencoder is regularized with a small UMAP graph-layout loss term (McInnes et al., 2018), which helps the very compact models learn both the local and the global structure of the manifold for each class.

To make accounting easier, we use the shorthand notation $\boldsymbol{x}^c$ to denote that feature $\boldsymbol{x}$ has label $c$, and $r^c$ to denote the autoencoder trained on $X^c$. Henceforth, we will refer to these autoencoders as *reconstructors*, as we care primarily about their ability to reconstruct features. The reconstruction

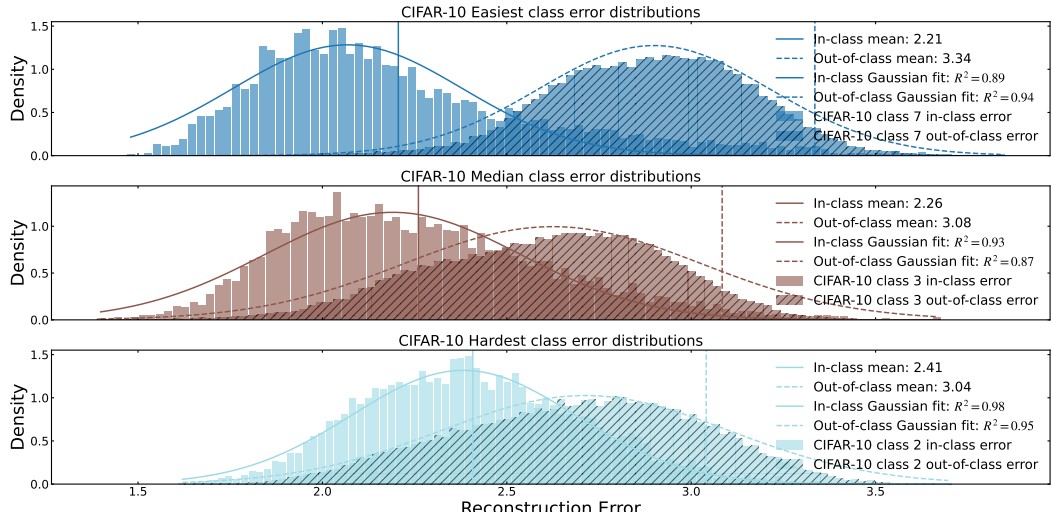

Figure 1: Reconstruction error distributions for in-class and out-of-class samples shown for the easiest, median, and hardest classes in the CIFAR-10 dataset, as measured by the average ratio of in-class and out-of-class reconstruction errors. In all cases, both in-class and out-of-class reconstruction errors are well-approximated with normal distributions. $R^2$ is the coefficient of determination, which is computed by evaluating the Gaussian fit curve at the center of each bin for 100-bin histograms. In-class refers to reconstruction error with the ground truth class's reconstructor; out-of-class refers to all other reconstruction errors.

error for a feature vector $\boldsymbol{x}$ with respect to reconstructor $r$ is defined as the difference between the original feature and the reconstruction.[1]

For most datasets with meaningful intra-class differences, we assume that on average the reconstructor trained on $X^c$ will be better at reconstructing features with label $c$ than features with other labels $c' \neq c$. Explicitly, letting $\Delta^{c'}(\boldsymbol{x}^c) = \|r^{c'}(\boldsymbol{x}^c) - \boldsymbol{x}^c\|$ denote the reconstruction error for a sample with label $c$ with respect to $r^{c'}$, we $\mathbb{E}_{X^c}[\Delta^c(\boldsymbol{x})] < \mathbb{E}_{X^{c'}}[\Delta^c(\boldsymbol{x})]$. We find this assumption to hold true in all experiments.

Moreover we find that for each reconstructor the in-class and out-of-class reconstruction errors tend to follow Gaussian distributions with distinct mean and variance. This is illustrated for three classes (the lowest, median, and highest average reconstruction error) from the CIFAR-10 dataset in Fig. 1.

The variance of these intra-class and inter-class reconstruction errors depends on the features used to fit the autoencoder, the complexity of the data, and the expressiveness of the encoder-decoder pair. Consequently, reconstruction errors can take on a wide range of values in $\mathbb{R}^+$, making it hard to draw conclusions from reconstruction errors alone. Reconstruction error ratios (RERs), on the other hand, produce dimensionless quantities $\phi_{12} = \Delta_1/\Delta_2$ of order one, which we can use to assess whether a new unlabeled sample belongs to class $c_1$ or $c_2$. Implementation details for RER computation are included in Appendix A.2. Autoencoders have seen moderate success when used for classification (Vincent et al., 2010), but have not reached the levels of state-of-the-art (SOTA) techniques. In the rest of this work, we show that the true power of RERs goes far beyond classification.

---

[1]Technically, this is the *magnitude* of the reconstruction error. For our purposes, the magnitude suffices, so we conflate the two terms.

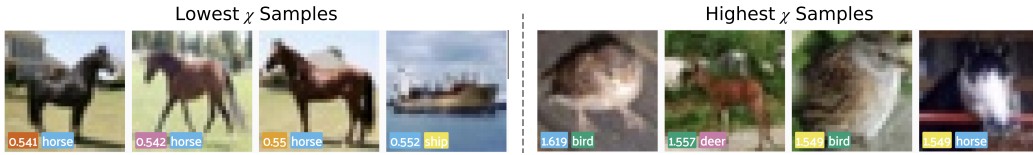

Figure 2: Visualization of $\chi$ for the easiest (left) and hardest (right) samples in CIFAR-10, using CLIP ViT-B/32 features used to train class reconstructors. Images generated using the Fiftyone library (Moore & Corso, 2020).

## 4 RERS AND CLASSIFICATION DIFFICULTY

### 4.1 RERS AS DATASET DETERMINANTS

Now we turn our attention to a specific reconstruction error ratio. Let

$$\chi(\boldsymbol{x}^c) = \frac{\Delta^c(\boldsymbol{x}^c)}{\min_{c' \neq c} \Delta^{c'}(\boldsymbol{x}^c)}, \tag{3}$$

be the ratio of the reconstruction error with ground truth class reconstructor to the minimum reconstruction error across all other reconstructors.

Intuitively, Eq. (3) probes the *classification difficulty* for sample $\boldsymbol{x}^c$ by comparing how *close* the sample is to its ground truth class manifold and how close it is to the closest alternative class. $\chi(\boldsymbol{x}^c) > 1$ indicates that there exists a class $c' \neq c$ whose reconstruction function represents the sample well relative to the ground truth class. $\chi(\boldsymbol{x}^c) < 1$, on the other hand, is a fairly strong indicator that the noisy ground truth class is accurate. Fig. 2 shows images from the four easiest (smallest $\chi$) and hardest (largest $\chi$) samples in CIFAR10. High-RER samples are often (but not always) located near class decision boundaries.

Computing $\chi$ for all samples and averaging over the entire dataset, we arrive at a dataset determinant,

$$\overline{\chi} = \mathbb{E}_{(\boldsymbol{X}, \boldsymbol{y})}[\chi(\boldsymbol{x}^c)], \tag{4}$$

which we interpret as the dataset's average classification difficulty. To validate $\overline{\chi}$ as a genuine measure of classification dataset difficulty, we systematically evaluate $\overline{\chi}$ on 19 visual datasets spanning more than 1.5 orders of magnitude in both the number of samples and the number of distinct classes. We then compare this value with the SOTA classification accuracy on the dataset obtained from PapersWithCode.[2] The results are summarized in Fig. 3, which showcases a strong relationship between $\overline{\chi}$ and the error rate ($1 - \text{Accuracy}$). We list all datasets utilized and detail our preprocessing steps in Appendix A.1. Fig. 12 in Appendix B.3 shows similar behavior for RERs on 10 out-of-domain medical datasets.

Quantitatively, when using the most expressive features (CLIP ViT-L/14), the Pearson correlation coefficient between $\overline{\chi}$ and the log-error-rate, $\log(1 - \text{Accuracy})$ is calculated to be $\rho = 0.640$. Oxford 102 Flowers is a significant outlier, which we believe may be due to differences in difficulty between the original train/val/test splits and the fact that our analysis is performed on a randomly selected subset. Removing this results in a substantially stronger correlation of $\rho = 0.781$. Additionally, datasets with many classes like ImageNet, SUN397, and Places205 notably drag the correlation down, which may be due to focus in the community on top-5 accuracy.

While specific values of $\overline{\chi}$ for a given dataset vary with the features used to train reconstructors, we find that the specific features used are immaterial. Figs. 10 and 11 as well as Table 3 in Appendix B.3 show the strong correlations between CLIP and DINOv2-style models, which are both strongly predictive of classification dataset difficulty. Pretrained ResNet-style models on the other hand are only weakly correlated with classification difficulty. We reiterate that once features have been generated, computing $\overline{\chi}$ takes seconds to minutes depending on the size of the dataset and the number of CPU cores available.

---

[2]For the DeepWeeds dataset no entry is listed on PapersWithCode so we instead use the highest accuracy reported in the DeepWeeds paper (Olsen et al., 2019).

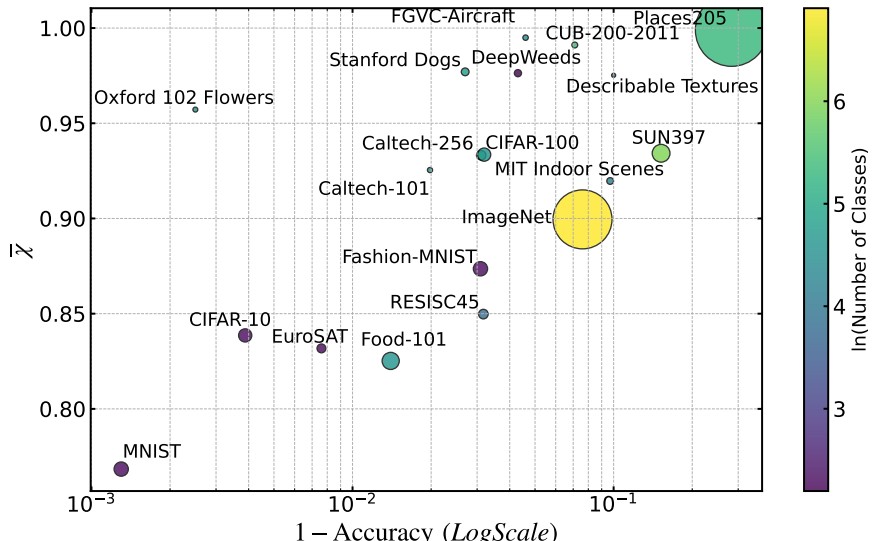

Figure 3: Scatterplot of SOTA classification error rate (plotted on a logarithmic scale) for 19 popular computer vision datasets versus estimated classification difficulty $\overline{\chi}$ computed using the reconstruction error ratio method. Autoencoders are trained on CLIP ViT-L/14 features and default parameters detailed in Table 1. Points are colored by the number of classes, scaled logarithmically, and are sized proportionally to the number of samples in the dataset. Log-error-rate and $\overline{\chi}$ are found to have a Pearson correlation coefficient of $\rho = 0.640$, and this increases to $\rho = 0.781$ when Oxford 102 Flowers is excluded.

## 4.2 FINITE SAMPLE SIZE CONTRIBUTIONS

RERs also provide a framework for decomposing classification difficulty. Ho & Basu (2002) argue that classification difficulty arises from three main sources: (1) Bayes error from class ambiguity, (2) decision boundary complexity, and (3) small sample size. RERs allow us to disentangle the first two from the latter. To our knowledge, this is the first time such a separation has been explicitly possible.

Because autoencoders are so fast and easy to train, we can see how $\overline{\chi}$ changes with the number of samples per class. For each dataset, we fit the reconstructor on a specified number of samples per class and then evaluate $\overline{\chi}$ across the entire dataset. Letting $\overline{\chi}_n$ denote the value $\overline{\chi}$ obtains for a given dataset when the reconstructors are fitted with $n$ examples per class, and let $\overline{\chi}_\infty$ denote the limit $n \to \infty$. Empirically, we find that for all datasets the data fit well to rational functions of the form:

$$\overline{\chi}_n = \frac{\overline{\chi}_\infty\, n^{\gamma_0} + \gamma_1}{n^{\gamma_0} + \gamma_2},$$

(5)

where $\gamma_0 = 1.808$ is fixed for all datasets. Fitting the 8 datasets that have at least 80 samples per class to this ansatz, we observe an average goodness of fit of $R^2 = 0.986$.[3] Specific parameter and $R^2$ values for each dataset are listed in Table 2. When restricting to datasets with $\overline{\chi} < 1$, all $R^2$ values exceed 0.99.

For datasets with 100 or more samples per class, this procedure gives us enough data points to robustly extrapolate to the infinite size limit. The results are shown in Fig. 4. Given $\overline{\chi}$ for the dataset as is, and an estimate for $\overline{\chi}_\infty$, we can estimate the contribution to classification difficulty arising from the finite size of the dataset as $\overline{\chi}_\infty - \overline{\chi}$.

---

[3]This ansatz only describes the data when finite size values do not cross 1. More delicate treatment is needed when $\overline{\chi}_n$ crosses 1. We leave this for future work.

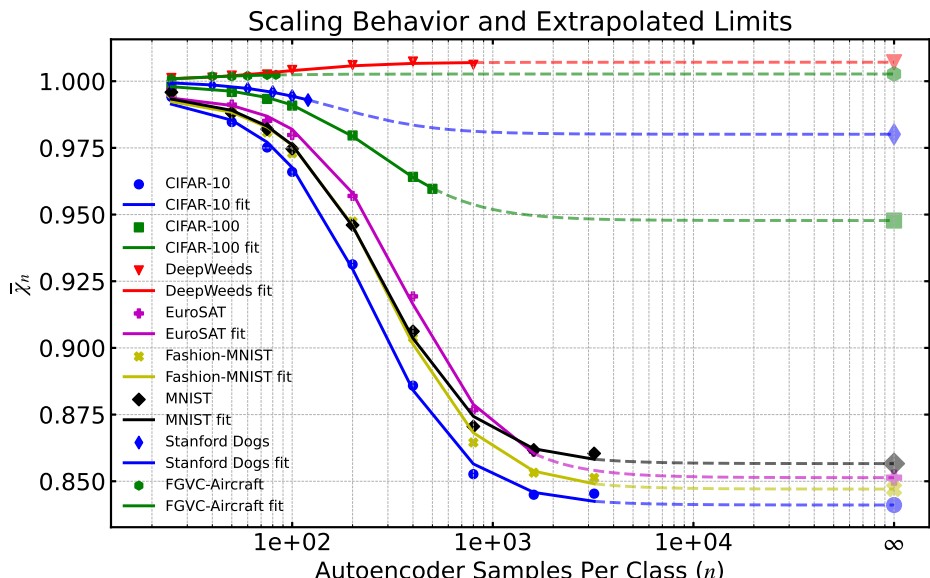

Figure 4: Dependence of dataset difficulty measure $\overline{\chi}$ (using CLIP ViT-B/32 features) on the number of samples per class used to train each reconstructor. We observe $\overline{\chi}_n$ to be well-behaved when $n \geq 20$, and for datasets where $\overline{\chi}_n$ does not oscillate around 1, scaling is well approximated by rational functions of the form (5). The infinite size limit extrapolated from this functional form is indicated by the large semi-transparent marker connected to the finite-size results by a dashed line.

### 4.3 LABEL NOISE AND BOUNDARY COMPLEXITY CONTRIBUTIONS

While on average we expect $\Delta^c(\boldsymbol{x}^c) < \Delta^c(\boldsymbol{x}^{c'})$, this will not always be the case. Our dataset may have epistemic uncertainty or ambiguously labeled samples, complex decision boundaries between classes, or even mislabeled samples. RERs provide a pathway to estimating these contributions to dataset difficulty as well.

In Appendix B.2 we show that $\overline{\chi}$ increases as a function of the noise in the dataset. Empirically, we verify this across all 19 datasets over a wide range of noise rates and types. Fig. 5 shows this dependence for symmetric, asymmetric, and confidence-based label noise.

We can make sense of these trends as follows: when we add a mistake via symmetric noise, we convert an example that almost certainly would not have had $\chi(\boldsymbol{x}) > 1$ instead of an example that almost certainly *will* have $\chi(\boldsymbol{x}) > 1$ so we add substantial error to the dataset. When we add confidence-based noise, we are converting examples near class decision boundaries into mistakes. On average, each confidence-based label mistake contributes less to the change in estimated noise. For asymmetric noise, transition matrix elements with nonzero entries are random, so at low noise rates we get the same behavior as symmetric noise. As we increase the amount of asymmetric noise, we significantly shift decision boundaries such that examples in asymmetrically connected classes become even more strongly tied together than confidence-based noise. As such, the contribution to estimated noise from asymmetric label mistakes decreases with the amount of noise added.

We can also use RERs to estimate the noise rate in the dataset. Let $\boldsymbol{x}^{\tilde{c}}$ denote that sample $\boldsymbol{x}$ has been assigned noisy label $\tilde{c}$, which may or not be $c$, and let $\Delta^{\tilde{c}}$ denote the reconstruction error obtained from reconstruction function $r^{\tilde{c}}$ trained on noisy samples $X^{\tilde{c}}$. This noise is assumed to include all sources of label noise and classification uncertainty in the dataset.

Letting $\Delta_{best}(\boldsymbol{x}) = \min_{c} \Delta^{\tilde{c}}(\boldsymbol{x})$ denote the minimum reconstruction error across all classes and

$$\Delta_{rand}(\boldsymbol{x}^{\tilde{c}}) = \mathbb{E}_{c' \in \mathcal{C} \setminus \{c\}}[\Delta_{\tilde{c}'}(\boldsymbol{x}^{\tilde{c}})] \tag{6}$$

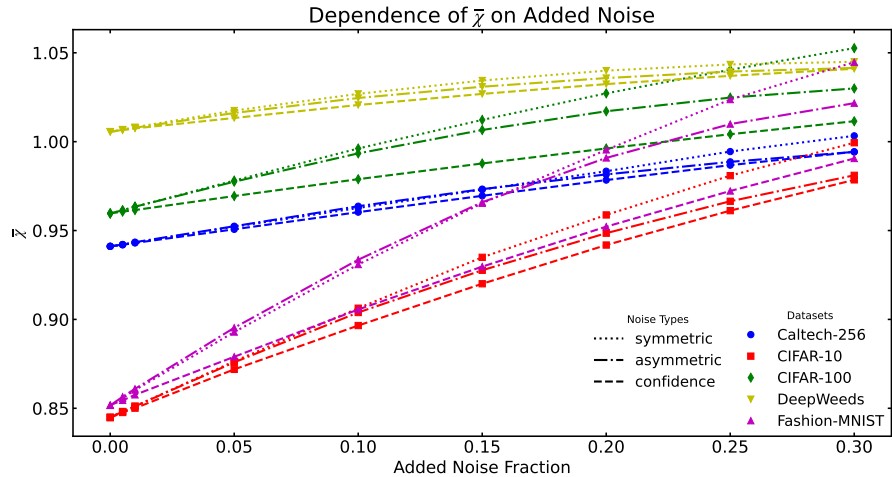

Figure 5: Relationship between $\overline{\chi}$ using CLIP ViT-B/32 features and symmetric, asymmetric, and confidence-based label noise for five exemplary datasets. Each point in the plot is generated by averaging over three random noise initializations.

denote the average reconstruction error obtained from a randomly chosen reconstructor, we can define the quantity

$$\chi_0 = \mathbb{E}_X\Big[\frac{\Delta_{\tilde{c}}(\boldsymbol{x}^{\tilde{c}})}{\Delta_{rand}(\boldsymbol{x}^{\tilde{c}})}\Big], \tag{7}$$

This gives us an approximation for the total noise:

$$\eta \approx \frac{\chi_0 - \chi_{rand}}{1 - \chi_{rand}}, \tag{8}$$

where $\chi_{rand} = \mathbb{E}_X[\Delta_{best}(\boldsymbol{x})/\Delta_{rand}(\boldsymbol{x})]$. The proof is included in Appendix B.2, along with empirical validation on multiple datasets.

### 4.4 APPLICATIONS

Curves of the form Eq. (5) allow us to estimate how adding a certain number of samples would impact the optimal classification accuracy we could achieve on the dataset. If classification accuracy across an entire dataset is known, finite-size contribution curves like those shown in Fig. 4 could be used to estimate the expected accuracy loss when randomly pruning $p\%$ of the data, allowing informed selection of prune rates that retain certain levels of performance. Conversely, these curves also permit estimating the performance boost from collecting or annotating a certain quantity of new data.

Finally, given $\overline{\chi}$ for a dataset $D$ and classification accuracy for a model trained on $D$, one can estimate how close to optimal the performance of that model is by plotting it on Fig. 3. Low accuracy scores paired with small $\overline{\chi}$ would indicate potential opportunity for improvement through preprocessing, model architecture, or training recipe.

## 5 RERS FOR MISLABEL DETECTION

Reconstruction error ratios also enable competitive mislabel detection through reinterpreting $\chi(\boldsymbol{x}^{\tilde{c}})$ as a *mistakenness* score for sample $\boldsymbol{x}$.

Consider the two possibilities: either the noisy label $\tilde{c}$ is correct ($\tilde{y} = y$) or it is incorrect ($\tilde{y} \neq y$).

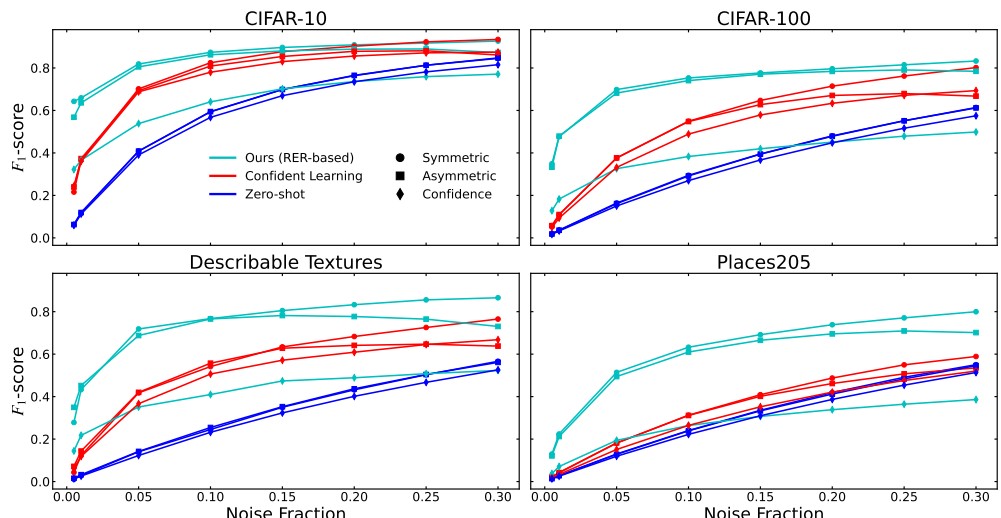

Figure 6: $F_1$-scores for Zero-Shot, Confident Learning, and RER-based mislabel detection methods on four exemplary datasets. RER thresholds are selected using ansatz (9). All three methods are compared using the same CLIP ViT-B/32 features. Each point represents an average over three noise initializations.

1. If $\tilde{c}$ is correct, then $\boldsymbol{x}^{\tilde{c}}$ will be in distribution for $X^{\tilde{c}}$, and the reconstruction error obtained by feeding $\boldsymbol{x}^{\tilde{c}}$ through $r^{\tilde{c}}$ will be small compared to the reconstruction error obtained with any other noisy class's reconstructor.

2. On the other hand, if $\tilde{c}$ is incorrect, there exists a class $c' \neq c$ such that $\boldsymbol{x}^{\tilde{c}}$ is in distribution for $X^{\tilde{c}'}$, and $\Delta^{\tilde{c}'}(\boldsymbol{x}^{\tilde{c}})$ will be small relative to $\Delta^{\tilde{c}}(\boldsymbol{x}^{\tilde{c}})$.

If we supplement these sample-wise mistakenness scores with a threshold, then we can assign a binary classification to each sample, specifying whether or not we believe its noisy label is a mistake. Denoting our threshold by $\chi^*$, we find that the simple ansatz

$$\hat{\chi}^* = \gamma_4 \chi_0^{\frac{-\gamma_5}{1+\gamma_6\eta}},\tag{9}$$

works remarkably well at generating binary mistake predictions with high $F_1$-scores. In Appendix C.1, we derive bounds on $\chi^*$ and show that this ansatz exhibits desirable scaling.

In practice, we find that this ansatz with the values $\gamma_4 = 1.01$, $\gamma_5 = 1.5$, $\gamma_6 = 13.8$ is close to optimal for symmetric and asymmetric noise outside of fine-grained classification scenarios. The ansatz tends to overshoot the optimal threshold for confidence-based and human annotator-based noise, but finds near-optimal thresholds for symmetric and asymmetric noise.

We test RER-based mislabel detection on four types of label noise: symmetric, asymmetric, confidence-based, and human annotator-based, defined as follows:

*Symmetric*: With probability $\eta$, a label $c_i$ is swapped uniformly where a label $c_j$, with $i \neq j$.

*Asymmetric*: With probability $\eta$, label $c_i$ is changed to $c_{i+1}$ modulo the number of classes.

*Confidence-Based*: A classifier is trained on the clean labels and used to run inference on the samples. For a given sample with label $c_i$, with probability $\eta$ the label is changed to the highest likelihood incorrect label predicted by the classifier for that sample.

*Human Annotator-Based*: A single human annotator assigns a label to each sample. This label is mistaken when it is in disagreement with the ground truth label resulting from aggregation and validation of human annotations. Mistakes from this set are randomly selected until $\eta$ (which must be less than or equal to the fraction of human annotator errors in the entire dataset) of the samples are assigned mistaken labels.

We compare RERs to the two best prior feature-based approaches: SimiFeat (Zhu et al., 2021) and a feature-based variant of Confident Learning (Srikanth et al., 2023a), as well as a zero-shot baseline, which we detail in Appendix A.3.2. We restrict ourselves to realistic noise regimes $0 \leq \eta \leq 0.3$, where at most 30% of labels are corrupted. We find that in this regime Confident Learning outperforms SimiFeat, and human annotator noise behaves nearly identically to confidence-based noise, so we omit these from plots for simplicity. Performance of RERs, Confident Learning, and zero-shot mislabel detection are shown in Fig. 6, where RER-based mislabel detection is found to consistently match or outperform all other feature-based methods under symmetry and asymmetric label noise when $\eta < 0.3$.

Taking threshold selection out of the equation, we also compute the area under the ROC curve (AUROC) for each dataset and noise setting, giving us a more complete picture of the strengths and weaknesses of each method. Illustrative AUROC curves for specific datasets are included in Fig. 13 in Appendix C.2. More generally, we find that RER-based mislabel detection consistently achieves higher AUROC scores for symmetric and asymmetric noise on hard datasets, which we define as datasets with SOTA classification accuracy $< 0.95$. Below we explain this by appealing to how Confident Learning and RERs work. We also note that AUROC scores obtained by RER-based mislabel detection are robust to the number of samples used to fit each reconstructor, stabilizing to near-optimal levels around 100 samples per class, as we demonstrate in Appendix C.2.

*Easy vs Hard Datasets:* Confident Learning trains a simple classifier and then assigns a label quality score based on the confidence of that classifier. If a classification dataset is easy, then even a simple classifier trained on rich features will be able to precisely learn class decision boundaries. RERs on the other hand train a separate reconstructor for each class. No reconstructor has explicit knowledge about other classes in the dataset. This makes the problem of mislabel detection more tractable by approximately decomposing it on a class-wise basis. For hard datasets, the tradeoff is well worth it, but for easy datasets the approximate decomposition may be substantial.

*(A)symmetric vs Confidence-Based Noise*: Reconstructors' lack of explicit interclass awareness also makes them especially susceptible to confidence-based noise, which perniciously persuade the reconstructions to learn class manifolds with slightly different shapes. Incorporating dataset-level awareness into the reconstructor training process is left for future work.

**Probabilistic Interpretation**: In addition to ranking samples according to their mistakenness and assigning binary clean/dirty labels, we also show in Appendix D.1 that RER mistakenness scores can be converted into mistakenness probabilities, reflecting consistent and accurate likelihoods that a given sample has a mistaken label. Furthermore, in Appendix D.2 we demonstrate that these probabilities are meaningful by way of a new metric which we call the confidence-weighted $F_1$-score. Given these probabilities, one could make more informed decisions about how many samples to send for reannotation to ensure a predetermined level of data quality on a fixed budget.

## 6 CONCLUSION

In this work, we introduced Reconstruction Error Ratios (RERs), a novel framework for analyzing classification datasets using class-wise autoencoders which we call reconstructors. This approach is fast, intuitive, interpretable, and model-agnostic, leveraging rich foundation model features and shallow autoencoders to enhance data curation and enable cross-dataset comparison.

Through a comprehensive analysis of 19 visual classification datasets varying in size and number of classes, we verify that RER-based dataset characteristics correlate strongly with SOTA classification model performance. Furthermore, we find that RER-based dataset difficulty behaves predictably as a function of the number of samples per class, providing useful information for dataset-reduction tasks like pruning an dataset-enhancement tasks like collection or annotation of unlabeled data. Subsequently, we demonstrate that RERs not only allow estimation of dataset-level noise rates, but also enable competitive detection of label mistakes. Along the way, we highlight applications in pruning, data collection, reannotation, and model selection.

While our current work focused on visual classification datasets, the principles underlying RERs are domain-independent. As such, the RER framework should be applicable to classification tasks in text, audio, time-series data, or even activity recognition. Future work will also extend RERs to derive dataset difficulty estimates for object detection or segmentation tasks.

## 7 REPRODUCIBILITY STATEMENT

All autoencoder and UMAP hyperparameters and training details, as well as data processing procedures, are documented in Appendix A. When testing mislabel detection methods, we use verified implementations of Confident Learning and SimiFeat from trusted open-source libraries. All mislabel detection experiments are run across three random noise settings with fixed random seeds. We will make our code publicly available along with the final paper.

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

## APPENDIX ROADMAP

The Appendix is organized as follows:

1. In Sec. A we detail of our technical implementation, including datasets used and preprocessing employed (A.1), training of autoencoders and Reconstruction Error Ratio computation hyperparameters (A.2), and detection of label mistakes (A.3.2).

2. Sec. B supplements our results on classification difficulty: in Sec. B.1 we document the observed finite sample size scaling behavior of reconstruction error ratios; Sec. B.2 details our theoretical estimation of dataset noise rates and validates this on visual classification datasets; Sec. B.3 shows the robustness of RER-based classification difficulty to specific feature backbone.

3. Sec. C.2 supplements our results on classification difficulty: in Sec. C.1 we derive bounds on and analyze the scaling properties of our threshold ansatz; Sec. C.2 provides additional details around our mislabel detection evaluation, as well as plots showing AUROC for specific datasets and AUROC averaged over all hard datasets.

4. Sec. D focuses on generating mistakenness probabilities from reconstruction error ratios. In Sec. D.1 we outline the protocol for turning RERs into probabilities and validate these probabilities in the context of the RER framework by comparing them to empirical mistake probabilities derived from added noise. Finally, Sec. D.2 argues that these probabilities are helpful by defining a new confidence-weighted $F_1$-score, proving its dependence on model confidence, and showing how RER-based and competitive mislabel detection methods fare with respect to this metric.

## A  IMPLEMENTATION DETAILS

### A.1  DATASETS

#### A.1.1  DATA DOMAINS

Our dataset classification difficulty experiments were run on 19 visual datasets spanning four visual task domains:

*Traditional Image Classification*: ImageNet Deng et al. (2009a), MNIST (LeCun et al., 2010), Fashion-MNIST (Xiao et al., 2017), CIFAR-10 and CIFAR-100 (Krizhevsky & Hinton, 2009),

Caltech-101 (Fei-Fei et al., 2004), Caltech-256 (Griffin et al., 2007), Describable Textures (Cimpoi et al., 2014), and DeepWeeds (Olsen et al., 2019).

*Fine-Grained Image Classification*: CUB-200-2011 (Wah et al., 2011), Stanford Dogs (Khosla et al., 2011), Oxford 102 Flowers (Nilsback & Zisserman, 2008), FGVC-Aircraft (Maji et al., 2013), and Food-101 (Bossard et al., 2014)

*Scene Recognition*: MIT Indoor Scenes (Quattoni & Torralba, 2009), Places205 (Zhou et al., 2017), and SUN397 (Xiao et al., 2010)

*Satellite Imagery*: EuroSAT (Helber et al., 2018; 2019) and RESISC45 (Cheng et al., 2017)

These datasets have state-of-the-art (SOTA) classification accuracies ranging from 71.7% (Places205) all the way up to 99.87% (MNIST). With the exception of DeepWeeds, SOTA classification accuracy used in dataset difficulty analyses was taken to be the top-ranking entry for each dataset's benchmark on PapersWithCode as of September 23, 2024.[4]

### A.1.2 Data Processing

MNIST, Fashion-MNIST, CIFAR-10, and CIFAR-100 were preserved as is. For all other datasets, we aggregated all samples and randomly generated $90/10$ train-test splits. In the case of Oxford 102 Flowers, which is the most significant outlier in our analyses, we hypothesize that significant differences in classification difficulty may have been present in the dataset's original splits.

The test set was used to validate the performance of classification models used to generate confidence-based noise. All analyses were performed exclusively on train splits. All non-PNG/JPG samples were discarded prior to embeddings generation.

### A.2 Reconstruction Error Ratio Computation

This section details UMAP and Autoencoder hyperparameters and training details.

*Training and hyperparameters*: Autoencoders with UMAP regularization loss are trained using the ParametricUMAP class from the umap-learn library (Sainburg et al., 2021). The encoder and decoder are defined in keras and each have one hidden layer. Small $l_2$-regularization and dropout are found to stabilize performance. ReLu activations are used for intermediate layers, and a sigmoid activation function is used after the last layer in the decoder. The number of training epochs is set to $n_{epochs} = 20$, but early stopping consistently occurs before that, as the loss converges quickly. Training is performed on CPU. Default hyperparameters used are detailed in Table 1. Systematic ablations lead us to the conclusion that variations in the number of components, dropout, regularization, and hidden layer dimension are largely inconsequential, resulting in no downstream performance changes beyond random chance. Aside from spread and min dist (detailed below), the most significant hyperparameter choices are the number of neighbors for UMAP and the relative weighting of the parametric reconstruction loss (relative to UMAP loss) in the autoencoder training. Hyperparameter sweeps for both are shown in Fig. 7, and in both cases, values obtained from RER-based difficulty estimation are found to robustly stabilize for sufficiently large hyperparamater values.

*Spread and min dist*: The only hyperparameters on which reconstruction error is found to depend strongly are the *spread* and *min dist*, which together control how tightly points are packed into the latent space. *min dist* is defined relative to *spread*, and we find that ratios close to one are near optimal. Intuitively, we believe that regularizing autoencoders using large spread and minimum distance between embedded points has a similar effect to KL-divergence in that it encourages exploration of the latent space. We note that the *spread* and *min dist* values that are found to work best result in negative values for UMAP's $a$ and $b$ force hyperparameters. As a result, the Python library throws warnings, but these do not hinder the resulting autoencoder's ability to represent in-distribution data. Using positive values of $a$ and $b$ results in a more well-behaved loss landscape but slightly diminished performance at mislabel detection. On the rare occasion that autoencoder training with *spread* $= 25$ and *min dist* $= 24$ threw an error, training was retried with *spread* $= 24$ and *min dist* $= 23$.

---

[4]For the DeepWeeds dataset, we use the highest classification accuracy reported in the original paper.

Table 1: Autoencoder Hyperparameters

| Hyperparameter | Value |
|---|---|
| Regularization Strength | 1e-6 |
| Dropout | 0.01 |
| Number of Components | 10 |
| Parametric Reconstruction Loss Weight | 20.0 |
| Batch Size | 64 |
| Hidden Dimensions | [256] |
| Number of Neighbors | 40 |
| Metric | Euclidean |
| Learning Rate | 0.1 |
| Repulsion Strength | 1.0 |
| Spread | 25.0 |
| Min Dist | 24.0 |

*Input features*: Unless explicitly noted, CLIP ViT-B/32 features are the inputs used to train our autoencoders. Before passing features into our autoencoders, we perform min-max normalization.

### A.3 MISLABEL DETECTION

In this subsection, we document all relevant implementation details involved in detecting label mistakes using reconstruction error ratios and the other methods used for comparison. For fair comparison, all mislabel detection methods are evaluated on the same fixed input features.

### A.3.1 LABEL NOISE GENERATION

Following (Srikanth et al., 2023a), four varieties of label noise were considered in this work: human annotator-based, symmetric, asymmetric, and confidence-based noise. Our implementations of symmetric and asymmetric label noise are adapted from the SimiFeat GitHub repo.

Human-annotator noise, which was only available for CIFAR-10 and CIFAR-100, was downloaded from https://github.com/UCSC-REAL/cifar-10-100n. These noisy labels contain $17.23\%$ and $40.20\%$ label errors respectively. To assess the performance of mislabel detection methods with varying amounts of human annotator noise, we isolated the indices where clean labels and hu-

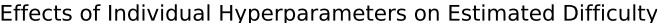
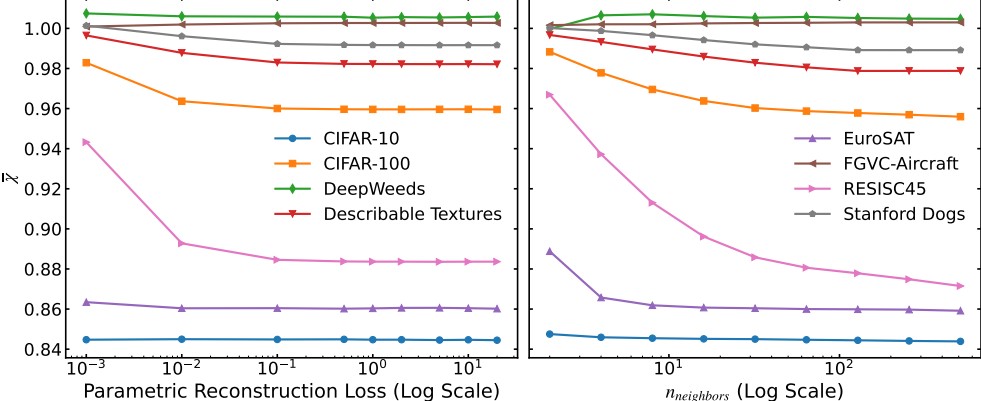

Figure 7: Hyperparameter sweeps for parametric reconstruction loss weight (left) and $n_{neighbors}$ (right) in class-wise autoencoder fitting for eight exemplary datasets. All non-specified hyperparameters are set to the defaults detailed in Table 1.

man annotator labels differed and randomly selected examples from this mistaken subset (without replacement) until we reached the desired noise rate.

Confidence-based noise was generated by training a classification model on the clean labels. For each sample, we take the highest-confidence incorrect prediction from our classifier: if the model's prediction is correct, we take its next highest-probability class. To retain consistency across datasets and avoid dataset-specific classifier architectures, we use the small and nano YOLOv8-cls classification models (Reis et al., 2024) from Ultralytics. In practice, we find that the relative performance of mislabel detection methods does not vary strongly with the specific classifier used to generate confidence-based noise.

### A.3.2 MISLABEL DETECTION METHODS

In our mislabel detection experiments, we compare our reconstruction error-based method to two three alternatives: (1) SimiFeat (Zhu et al., 2021), (2) Confident Learning (Northcutt et al., 2021) , and (3) a zero-shot baseline. All methods are compared using the same features. In practice, we find that Confident Learning consistently matches or outperforms SimiFeat, so we omit SimiFeat from plots for simplicity.

*Zero-Shot Mislabel Detection*: All class names were tokenized and embedded with the standard CLIP ViT-B/32 text encoder with the template *"A photo of a $\langle class\_name \rangle$"*. The normalized sample (image) features are multiplied by these normalized class name embeddings to produce logits, following OpenAI's original recipe, with the largest logit corresponding to the predicted label. Logits are converted to probabilities via the softmax. From there, the mistakenness method from the FiftyOne Brain library is used, as described below.

For a given sample, let $p_i$ be the probability associated with class $c_i$. Furthermore, let $m$ modulate whether the predicted label agrees with the supposed ground truth label:

$$m = \begin{cases} 1, & \text{if} \quad \hat{y} = y \\ -1, & \text{otherwise} \end{cases} \tag{10}$$

The mistakenness for a sample $(\boldsymbol{x}, y)$ is defined as:

$$mistakenness = \frac{1 + m * e^{\sum p_i \log p_i}}{2}, \tag{11}$$

which is in the range $[0, 1]$, with higher values indicating highly-confident misalignment with the ground truth label. A symmetric threshold of $0.5$ is used in all experiments.

*SimiFeat*: We use the implementation of SimiFeat in the docta.ai library (Zhu et al., 2023). All configuration hyperparameters are used as is from the docta.ai examples, including the selection cutoff at $0.2$.

*Confident Learning*: We use the implementation of Confident Learning from the cleanlab Python library. Following the recipe outlined in (Srikanth et al., 2023a), we use a simple logistic regression classifier with $max\_iter = 1000$. For all other hyperparameters, Cleanlab's defaults are used in all experiments.

## B ADDITIONAL CLASSIFICATION DIFFICULTY RESULTS

### B.1 RECONSTRUCTION ERROR RATIOS AND FINITE SAMPLE SIZE

In Sec. 4.2, we show that $\overline{\chi}$ as a number of samples per class can be fitted well to Eq. (5). The results of fitting to this functional form are detailed in Table 2.

We also observe that other reconstruction error ratios such as $\chi_0$ and $\chi_{rand}$ obey the same scaling, with the same exponent, as illustrated in Fig. 8.

We stress that the infinite-size scaling analyses shown in Fig. 4 and Fig. 8 are not necessary for estimating the classification difficulty of the dataset as a whole or for detecting label mistakes.

Table 2: Fitting parameters and $R^2$ values for Eq. (5) for 8 datasets with at least 80 samples per class using CLIP ViT-B/32 features.

| Dataset | $\overline{\chi}_\infty$ | $\gamma_1$ | $\gamma_2$ | $R^2$ |
|---------|-----------|-----------|-----------|-------|
| **CIFAR-10** | 0.8411 | 19755.34 | 19875.82 | 0.9986 |
| **CIFAR-100** | 0.9478 | 23629.23 | 23660.84 | 0.9993 |
| **DeepWeeds** | 1.0071 | 3580.11 | 3578.83 | 0.9562 |
| **EuroSAT** | 0.8513 | 41763.83 | 41981.35 | 0.9984 |
| **Fashion-MNIST** | 0.8471 | 29558.87 | 29738.47 | 0.9983 |
| **MNIST** | 0.8566 | 25904.07 | 26032.33 | 0.9986 |
| **Stanford Dogs** | 0.9801 | 10564.68 | 10565.00 | 0.9986 |
| **FGVC-Aircraft** | 1.0027 | 290.81 | 291.17 | 0.9429 |
| **Mean** | - | - | - | 0.9864 |

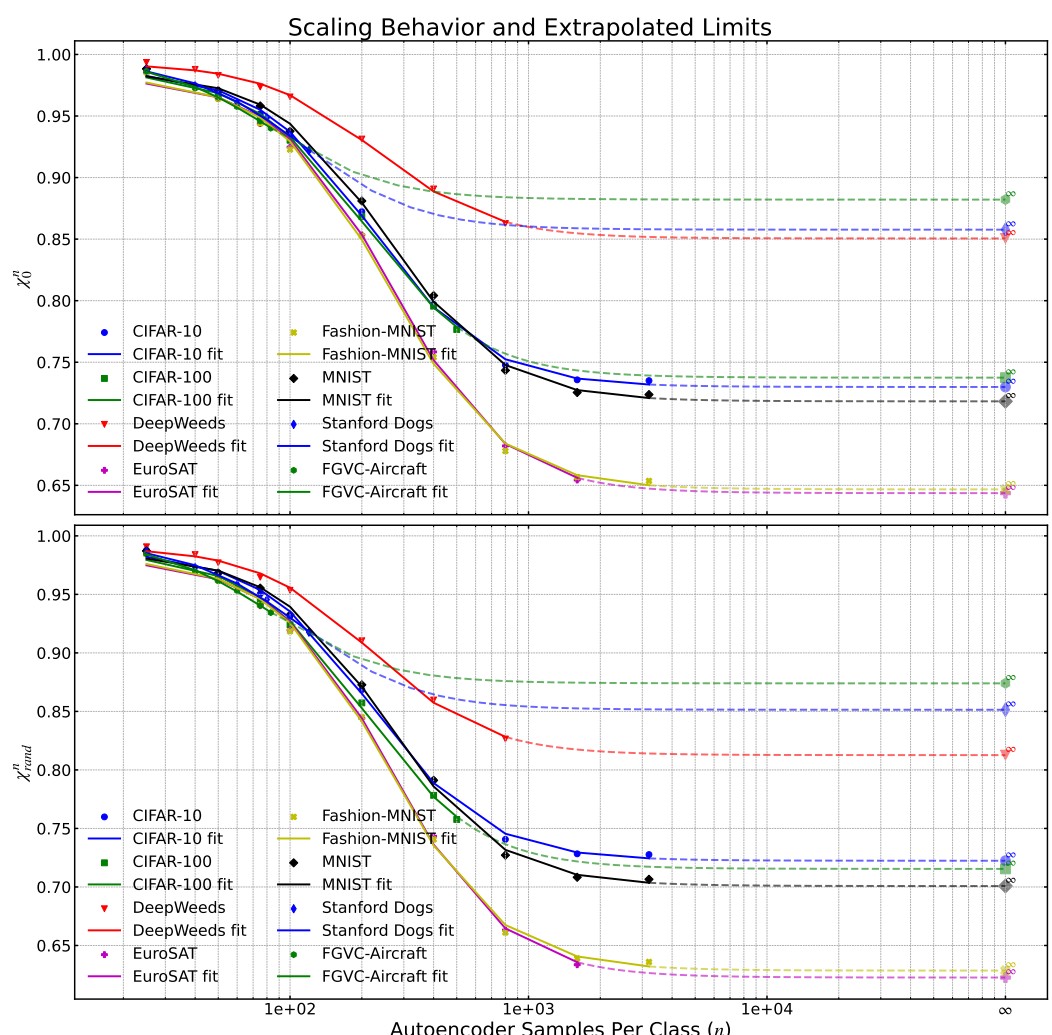

Figure 8: Finite-size scaling behavior of $\chi_0$ and $\chi_{rand}$ using CLIP ViT-B/32 features and fitting to equations of the form Eq. (5) with exponent 1.808.

## B.2 ESTIMATING THE NOISE RATE IN THE DATASET

Let $x^{\tilde{c}}$ denote that sample $x$ has been assigned noisy label $\tilde{c}$, which may or not be $c$, and let $\Delta^{\tilde{c}}$ denote the reconstruction error obtained from reconstruction function $r^{\tilde{c}}$ trained on noisy samples

$X^{\tilde{c}}$. This noise is assumed to include all sources of label noise and classification uncertainty in the dataset.

First, we will show that $\overline{\chi}$ increases with noise:

Consider

$$\chi(\boldsymbol{x}^c) = \frac{\Delta^c(\boldsymbol{x}^c)}{\min_{c' \neq c} \Delta^{c'}(\boldsymbol{x}^c)}, \tag{12}$$

With probability $\eta$ there is an error. In this case, $\min_{c' \neq c} \Delta^{c'}(\boldsymbol{x}^c) = \Delta_{best}(\boldsymbol{x}^c)$ and $\Delta^c(\boldsymbol{x}^c) = \Delta_{other}(\boldsymbol{x}^c)$, where $\Delta_{other}$ can be the reconstruction error with any other class than the clean ground truth class. With probability $1 - \eta$ the label is clean, and $\chi$ resolves to $\Delta_{best}/\Delta_2$, where $\Delta_2$ is the second lowest reconstruction error.

By linearity of expectation values,

$$\overline{\chi} = (1 - \eta) \, \mathbb{E}_X \big[ \Delta_{best}/\Delta_2 \big] + \eta \, \mathbb{E}_X \big[ \Delta_{other}/\Delta_{best} \big]. \tag{13}$$

Rearranging and noting that $\Delta_{best}/\Delta_2 < 1$ and $\Delta_{other}/\Delta_{best} > 1$, we arrive at

$$\overline{\chi} = \mathbb{E}_X \big[ \Delta_{best}/\Delta_2 \big] + \eta \, \mathbb{E}_X \big[ \Delta_{other}/\Delta_{best} - \Delta_{best}/\Delta_2 \big], \tag{14}$$

which increases monotonically with $\eta$.

We do not know $\Delta_{other}$, so we cannot explicitly evaluate $\eta$ from this equation. However, we can estimate $\eta$ from $\chi_0$, also reproduced here for clarity:

$$\chi_0 = \mathbb{E}_X \Big[ \frac{\Delta_{\tilde{c}}(\boldsymbol{x}^{\tilde{c}})}{\Delta_{rand}(\boldsymbol{x}^{\tilde{c}})} \Big], \tag{15}$$

To first order, with probability $\eta$, there is some sort of mistake and $\tilde{c} \neq c$. By linearity, $\chi_0$ decomposes into:

$$\chi_0 = (1 - \eta) \, \mathbb{E}_X \Big[ \frac{\Delta_{\tilde{c}}(\boldsymbol{x}^c)}{\Delta_{rand}(\boldsymbol{x}^c)} \Big] + \eta \, \mathbb{E}_X \Big[ \frac{\Delta_{\tilde{c}}(\boldsymbol{x}^{c'})}{\Delta_{rand}(\boldsymbol{x}^{c'})} \Big], \tag{16}$$

where $c' \neq c$. The numerator in the second term can be identified as (6), so the second expectation value in (16) resolves to the identity and the equation simplifies to

$$\chi_0 = (1 - \eta) \, \mathbb{E}_X \Big[ \frac{\Delta_{\tilde{c}}(\boldsymbol{x}^c)}{\Delta_{rand}(\boldsymbol{x}^c)} \Big] + \eta. \tag{17}$$

In the limit $\eta \ll 1$, when noise is symmetrically distributed across spurious classes, we can approximate $\Delta_{\tilde{c}}(\boldsymbol{x}^c) \approx \min \boldsymbol{\Delta}(\boldsymbol{x}^c)$. In other words, if the noise is small enough, our reconstruction function trained on noisy class $\tilde{c}$ will generate the smallest reconstruction errors (among all noisy class reconstruction functions) for features that belong in class $c$. We will refer to this minimum as $\Delta_{best}(\boldsymbol{x})$.

Employing this approximation and denoting

$$\chi_{rand} = \mathbb{E}_X[\Delta_{best}(\boldsymbol{x})/\Delta_{rand}(\boldsymbol{x})], \tag{18}$$

we arrive at

$$\chi_0 \approx (1 - \eta)\chi_{rand} + \eta. \tag{19}$$

Note that we can explicitly compute both (7) and (18) from our noisy data, so that rearranging (19), we can estimate the noise rate in the dataset as:

$$\eta \approx \frac{\chi_0 - \chi_{rand}}{1 - \chi_{rand}}. \tag{20}$$

Figure 9 showcases the predictive power of Eq. (20) for nine datasets across symmetric, asymmetric and confidence-based label noise. We first estimate the *intrinsic* noise in the dataset. We then add $\eta_{added}$ label noise and estimate the total noise in the corrupted dataset. The dashed line with unit slope and intercept $\eta_{intrinsic}$ charts the ideal performance of Eq. (20) as a function of the added label noise.

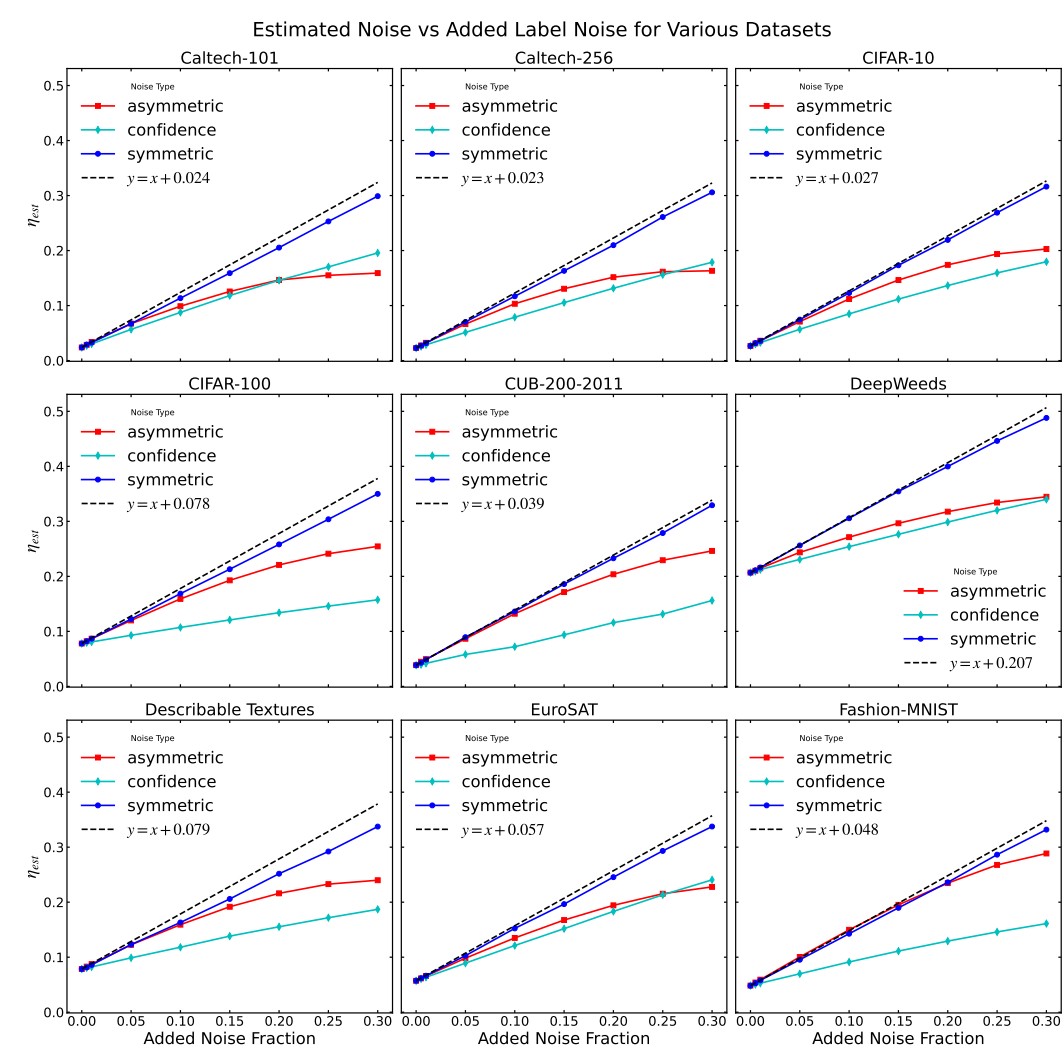

Figure 9: Estimated noise in the dataset versus noise added for 9 exemplary datasets.

### B.3 ROBUSTNESS ANALYSIS

All results in the body of the paper utilize features generated from either CLIP ViT-B/32 or CLIP ViT-$L/14$ vision encoders. However, the reconstruction error ratio framework is not specific to CLIP-style models. We demonstrate this explicitly by computing $\overline{\chi}$ for all 18 non-ImageNet datasets from the main text using features from five models. We report the correlations between the dataset difficulty scores estimated with these five sets of features in Fig. 10. Note that we exclude ImageNet from this analysis, as the ResNet model we probe was pretrained on ImageNet, which could lead to unfair comparison. Given the generality of our findings, we also expect that RERs are intimately related to classification margins, among the varied signals that are captured by the RER framework. We plan to formalize this connection in future work.

Beyond correlation on the dataset level, we find that various CLIP and DINOv2 backbones produce reconstruction error ratios that align well on the class and sample levels. Concretely, we analyze how consistent the rankings are across features by computing the Spearman Rank correlation and the normalized discounted cumulative gain (nDCG). Both metrics are computed on the sample level by taking the $\chi(\boldsymbol{x})$ for each sample, and are computed on the class level by taking the average $\chi$ value across all samples with a specific ground truth label, $\chi_c = \mathbb{E}_{\boldsymbol{x} \in X^c}[\chi(x)]$. We choose the Spearman Rank correlation rather than the Kendall $\tau$ because the latter depends strongly on the

number of elements in the set to be ranked, leading to values that vary widely from dataset to dataset based on the number of samples and the number of classes. The results for CIFAR10 are shown in the first and second heatmaps in Fig. 11, demonstrating moderate-to-strong correlation between features.

We believe that these rank correlations alone undersell the effective alignment in RER ordering between features, as in practice, the most important samples for mislabel detection are the highest-scoring samples. To draw out this aspect, we look at the nDCG, which gives more weight to higher scoring samples (elements at the top of the ranking). Before computing the nDCG, we perform min-max normalization on the scores generated by each feature backbone. For both sample-wise (third heatmap) and class-wise ordering (fourth heatmap) in Fig. 11, we see very similar rankings across all CLIP and DINOv2 models. We also perform this same analysis on all 19 datasets in our study and present the results for CLIP ViT-B/32 ↔ DINOv2-B in Table 3, underlining the generality of this finding.

In addition to the RER framework's robustness to a specific feature backbone, the framework is remarkably robust to out-of-domain datasets. While foundation models like CLIP and DINOv2 were likely trained primarily on natural images, we observe that the feature extraction capabilities of both models are strong enough to accommodate medical imagery. Without any modification to our procedures, we apply RERs for classification difficulty assessment on the 10 datasets in MedM-NISTv2 Yang et al. (2023) which feature 2D images and are designed for non-binary classification tasks. There is no definitive source for SOTA classification accuracies for these medical datasets, so in Fig. 12 we plot the estimated classification difficulty against the log-error rate of the best-performing method listed for each dataset in the MedMNISTv2 paper.

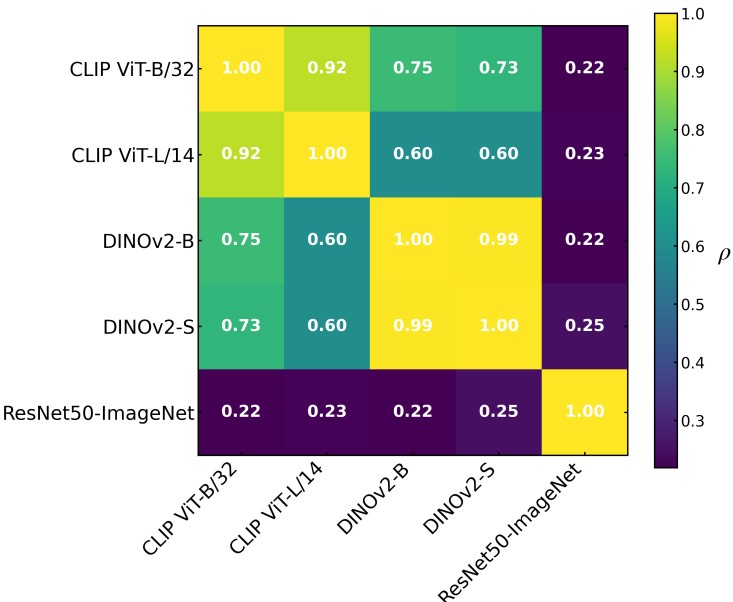

Figure 10: Correlations among $\overline{\chi}$ values generated for all 18 non-ImageNet datasets considered when training autoencoders on various features. Within model families, (CLIP ViT-B/32 ↔ CLIP ViT-L/14 and DINOv2-B ↔ DINOv2-S) there is strong positive correlation. CLIP and DINOv2-style models are expressive enough that they exhibit relatively strong inter-family correlation. Pre-trained ResNet50 features are found to correlate only weakly with other features and with SOTA log-error-rate across datasets.

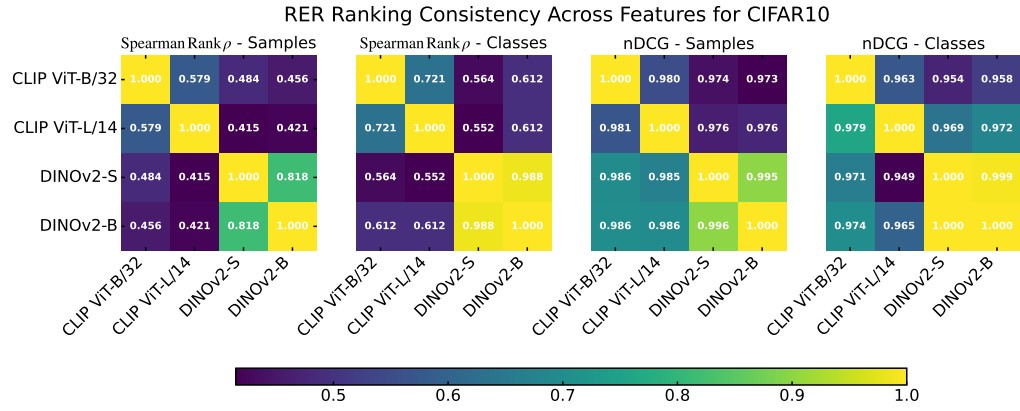

Figure 11: Alignment in ordering of $\chi(\boldsymbol{x})$ across multiple CLIP and DINOv2 models. Spearman Rank Correlation (first and second plots) characterizes the overall quality of the alignment, whereas nDCG (third and fourth plots) more heavily weights the top portion of the ranking.

## C    ADDITIONAL MISLABEL DETECTION RESULTS

### C.1    CHOOSING A THRESHOLD FOR MISLABEL DETECTION

Letting $\boldsymbol{y}^{pred}$ be our vector of mislabel predictions, we classify a sample as mislabeled when the RER is above a fixed threshold:

$$\boldsymbol{y}_j^{pred} = \begin{cases} 0, & \text{if } \chi(\boldsymbol{X}_{j,:}^{\tilde{c}}) < \chi_{thresh} \\ 1, & \text{otherwise} \end{cases} \tag{21}$$

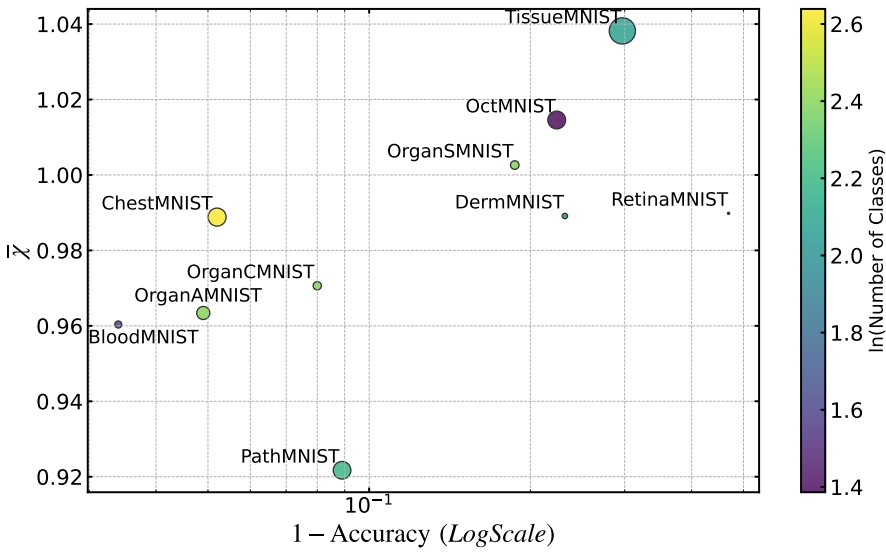

Figure 12: Scatterplot of best classification error rate from MedMNISTv2 paper Yang et al. (2023) (plotted on a logarithmic scale) for all 10 2D datasets in MedMNISTv2 with more than 2 classes versus estimated classification difficulty $\overline{\chi}$ computed using the reconstruction error ratio method. Autoencoders are trained on CLIP ViT-L/14 features and default parameters detailed in Table 1. Points are colored by the number of classes, scaled logarithmically, and are sized proportionately to the number of samples in the dataset.

Table 3: RER Ordering Alignment Between CLIP ViT-B/32 and DINOv2-B

| Dataset | Spearman Rank $\rho$ | | nDCG | |
|---|---|---|---|---|
| | Classes | Samples | Classes | Samples |
| Caltech-101 | 0.667 | 0.701 | 0.979 | 0.991 |
| Caltech-256 | 0.748 | 0.611 | 0.992 | 0.992 |
| CIFAR-10 | 0.612 | 0.456 | 0.958 | 0.973 |
| CIFAR-100 | 0.815 | 0.526 | 0.988 | 0.980 |
| CUB-200-2011 | 0.444 | 0.295 | 0.961 | 0.974 |
| DeepWeeds | 0.867 | 0.521 | 0.991 | 0.975 |
| Describable Textures | 0.830 | 0.702 | 0.973 | 0.984 |
| EuroSAT | 0.721 | 0.253 | 0.981 | 0.954 |
| Fashion-MNIST | 0.988 | 0.711 | 1.000 | 0.981 |
| FGVC-Aircraft | 0.630 | 0.259 | 0.987 | 0.969 |
| Food-101 | 0.597 | 0.295 | 0.977 | 0.977 |
| ImageNet | 0.675 | 0.481 | 0.987 | 0.990 |
| MIT Indoor Scenes | 0.548 | 0.209 | 0.923 | 0.966 |
| MNIST | 0.661 | 0.545 | 0.947 | 0.980 |
| Oxford 102 Flowers | 0.745 | 0.603 | 0.983 | 0.984 |
| Places205 | 0.774 | 0.514 | 0.990 | 0.982 |
| RESISC45 | 0.709 | 0.483 | 0.957 | 0.974 |
| Stanford Dogs | 0.666 | 0.354 | 0.973 | 0.980 |
| SUN397 | 0.298 | 0.117 | 0.939 | 0.975 |

Our goal is to select the threshold $\chi^*$ which maximizes our $F_1$ score:

$$\chi^* = \underset{\chi_{thresh}}{\arg\min} F_1(\chi_{thresh}), \tag{22}$$

We cannot compute $\chi^*$ exactly from our noisy dataset using this framework as the $F_1$-score threshold is not an intrinsic attribute of a dataset. However, we can derive some heuristic bounds and estimate this threshold from the data.

In the ideal scenario of minimal noise and sufficiently well-behaved data, $\Delta_{\tilde{c}}(\boldsymbol{x}^c) = \min \boldsymbol{\Delta}(\boldsymbol{x}^c)$. Thus, when $\tilde{c} = c$, $\chi_{clean} = \mathbb{E}_X[\chi(\boldsymbol{x}^{c=\tilde{c}})] < 1$. On the other hand, when $\tilde{c} \neq c$, on average $\chi_{dirty} = \mathbb{E}_X[\chi(\boldsymbol{x}^{c\neq\tilde{c}})] = \chi_{rand}^{-1} > 1$. Our threshold should be able to distinguish between these two scenarios, so

$$\chi_{clean} \leq \chi^* \leq \chi_{dirty}. \tag{23}$$

As we increase the noise rate in the dataset, the preferential ability of $r^{\tilde{c}}$ to reconstruct samples with clean label $c$ diminishes. At some critical noise rate $\eta_{crit}$, which depends on the type of noise, $r^{\tilde{c}}$ will no longer be better at reconstructing a sample $\boldsymbol{x}^c$ than another reconstruction function $r^{\tilde{c}'}$. Using the superscript $\eta$ to indicate the dependence on noise rate, we have that

$$\lim_{\eta \to \eta_{crit}} \chi_{clean}^{\eta} = \chi_{dirty}^{\eta} = 1. \tag{24}$$

By the squeeze theorem, this implies that $\lim_{\eta \to \eta_{crit}} \chi^* = 1$. Now consider how $\chi^*$ depends on the noise rate. By definition, $\chi^*$ is the threshold that maximizes our $F_1$-score.

At low noise rates $\eta \ll 1$, the $F_1$-score is more susceptible to false positives than to false negatives, and $F_1(\chi_{thresh})$ is maximized by setting a high threshold, whereas for high error rates it is best to set the threshold on the lower side. Thus, we expect $\chi^*$ to monotonically decrease with $\eta$, which implies

$$1 \leq \chi^* \leq \chi_{rand}^{-1},[5] \tag{25}$$

Furthermore, the rate of change in $\chi^*$ should be higher for smaller $\eta$. While $\chi_{rand}^{-1}$ decreases with $\eta$, the rate at which $\chi_{rand}^{-1}$ approaches unity is not guaranteed to coincide with the rate at which $r^*$ approaches unity. Nevertheless, we can construct an ansatz that has the desired properties.

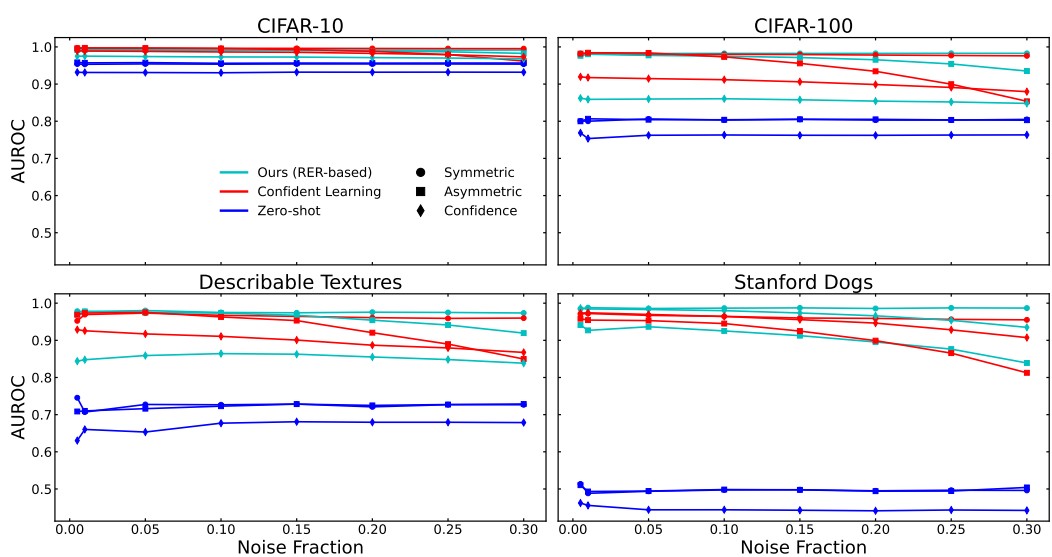

Figure 13: AUROC scores for binary mislabel detection tasks on four datasets.

Consider the quantity $\chi_0$ that we previously introduced, reproduced here for clarity:

$$\chi_0 = (1 - \eta)\chi_{rand} + \eta, \tag{26}$$

As $0 \le \eta \le 1$ and $\chi_{rand} \le 1$, we also have $1 \le \chi_0^{-1} \le \chi_{rand}^{-1}$. Differentiating with respect to $\eta$,

$$\frac{d\chi_0}{d\eta} = (1 - \eta)\frac{d\chi_{rand}}{d\eta} + (1 - \chi_{rand}), \tag{27}$$

and observing that $(1 - \eta) > 0$, $\frac{d\chi_{rand}}{d\eta} > 0$, and $(1 - \chi_{rand}) > 0$, $\frac{d\chi_0}{d\eta} > 0$, implying that $\chi_0^{-1}$ decreases with $\eta$. Additionally, the rate of change in $\chi_{rand}^{-1}$ decreases with $\eta$.

## C.2 EVALUATING RER-BASED MISLABEL DETECTION

Employing the RER threshold ansatz Eq. (9), we find that in almost all noise regimes and on almost all datasets, RER-based mislabel detection produces higher $F_1$ scores than competitive feature-based methods such as SimiFeat and Confident Learning (with a logistic regression classifier) for symmetric and asymmetric noise. However, Eq. (9) is not a fundamental element of the RER mislabel detection. We turn to AUROC to remove threshold selection from the equation.

Fig. 13 shows AUROC scores for four datasets, where we generally observe that same trends as with $F_1$-scores: RER-based mislabel detection excels under symmetric noise, typically outperforms competitive methods under asymmetric noise, and sits somewhere in between zero-shot and state-of-the-art approaches for confidence-based noise.

We can gain even deeper insight when we stratify our datasets into *easy* (SOTA accuracy $> 0.95$) and *hard* (SOTA accuracy $> 0.95$). On harder datasets, we consistently outperform Confident Learning by a wide margin on symmetric and asymmetric label noise, but still fall short in confidence-based noise scenarios, as shown in Fig. 14.

We find that for datasets with SOTA classification accuracy below $95\%$, the AUROC obtained from reconstruction error ratios is on average higher than Confident Learning's AUROC for both symmetric and asymmetric noise.

We also find that RER-based mislabel detection performance converges rapidly in the number of samples used to fit the class reconstructors. As we highlight in Fig. 15 for CIFAR-10 and CIFAR-100, RER-based mislabel detection AUROC stabilizes when $\sim 100$ are used to fit each reconstructor.

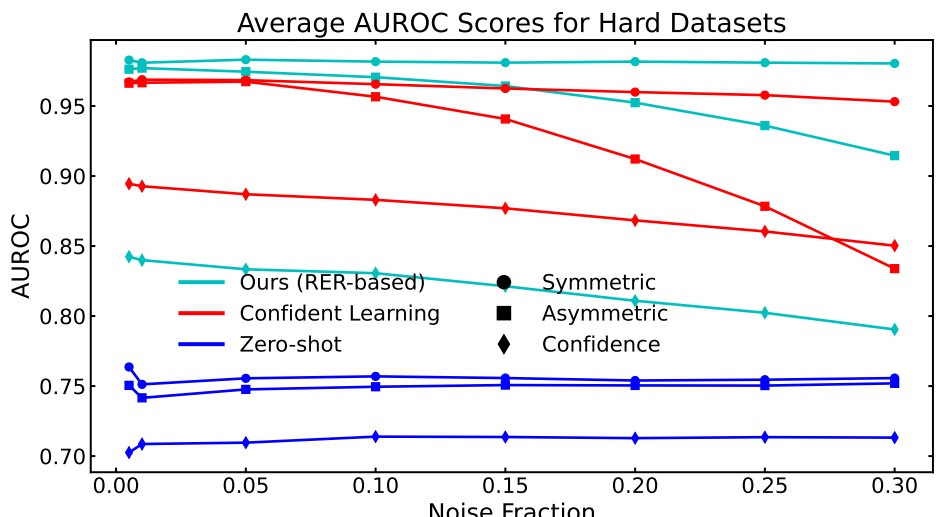

Figure 14: Average AUROC across all 5 hard classification datasets (which have SOTA classification accuracy < 0.95).

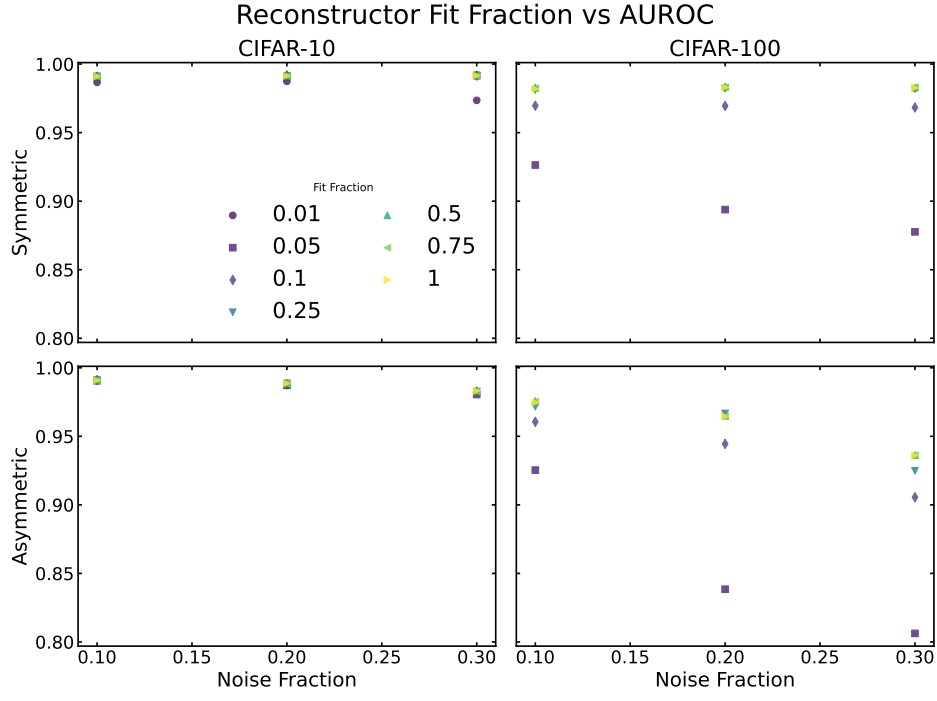

Figure 15: AUROC scores for CIFAR-10 and CIFAR-100 with reconstructors fitted using a fraction of the entire dataset. CLIP ViT-B/32 features are used in all cases.

# D    RECONSTRUCTION ERROR RATIOS AND THE LIKELIHOOD OF A LABEL MISTAKE

## D.1    TURNING RERS INTO PROBABILITIES

Beyond having an estimated threshold $\chi^*$ at which to classify something as mislabeled, it would be ideal to assign a probability to each sample describing the likelihood that said sample is mislabeled. Concretely, we aim to obtain $p(\text{mistake}|\chi)$, the probability that a sample has an erroneous label given that it registered an RER of $\chi$.

We can estimate this probability distribution using Bayes' Theorem, inverting the problem as:

$$p(\text{mistake}|\chi) = \frac{p(\chi|\text{mistake})p(\text{mistake})}{p(\chi)}, \tag{28}$$

The denominator on the right hand side of (28) can be estimated from the RERs across our dataset, $\{\chi(x_j^{\tilde{c}})|x_j^{\tilde{c}} \in X\}$, which we have already computed. The mistake probability across the dataset can be estimated by inverting (7) to obtain

$$p(\text{mistake}) = \eta = \frac{\chi_0 - \chi_{rand}}{1 - \chi_{rand}}, \tag{29}$$

where both $\chi_0$ and $\chi_{rand}$ can be computed explicitly from $\boldsymbol{\Delta}$ and $\{\tilde{y}_j\}_{j\in[N]}$.

The final piece of the puzzle is approximating the distribution of mislabeled RERs. Fortunately, we can estimate this distribution by *emulating* the creation of errors in the dataset as follows:

For each sample $x_j^{\tilde{c}}$ with noisy label class $c$, randomly flip its class to some other class $c'$. Then construct the ratio:

$$\frac{\Delta^{\tilde{c}'}(x_j^{\tilde{c}})}{\min_{c''\neq c'} \Delta^{\tilde{c}''}(x_j^{\tilde{c}})}. \tag{30}$$

Even if $x_j^{\tilde{c}}$ was already mislabeled, it will also be an error after this label swapping procedure with probability $\frac{N_c-1}{N_c}$, so this procedure successfully generates mistakes with probability $(1 - \eta) + \eta * \frac{N_c-1}{N_c}$.

In practice, emulating errors amounts to picking elements from $\boldsymbol{\Delta}$ in a certain way. A slight difference between this emulation and real mistakes is that noisy labels were used to fit the noisy autoencoder for each class, which was then used to construct the RERs, whereas in this scenario the emulated errors do not influence autoencoder fitting. Nevertheless, this approach works remarkably well, as we illustrate for CIFAR-10, CIFAR-100, and the Stanford Dogs dataset in Fig. 16.

To estimate the mistakenness posterior, we use kernel density estimation with reflection at the right boundary to approximate $p(\chi_x)$ and $p(\chi_x|\text{mistake})$ from finite sample populations.

At low rates of added noise $0.01 \leq \eta_{added} \leq 0.05$, our posterior overestimates compared to the empirically computed likelihood because intrinsic label noise, which we do not account for in our empirical estimates contributes non-negligibly.

## D.2    VALIDATING THE PROBABILITIES

Following this procedure and applying Bayes' Theorem, we arrive at probabilities for each sample which tell us how likely it is, given the sample's RER, that its label is erroneous. The probability density functions estimated with this method align remarkably well with true mistakenness probabilities, which we compute by comparing the noisy and clean labels and binning by RER. However, this does not necessarily imply that our probabilities are meaningful in a broader sense. In particular, we may ask how much is gained by assigning said probabilities over a binary mask exclusively predicting whether or not each sample is mistaken.

We propose to evaluate the *helpfulness* of a set of probabilities with a new metric, which we define below.

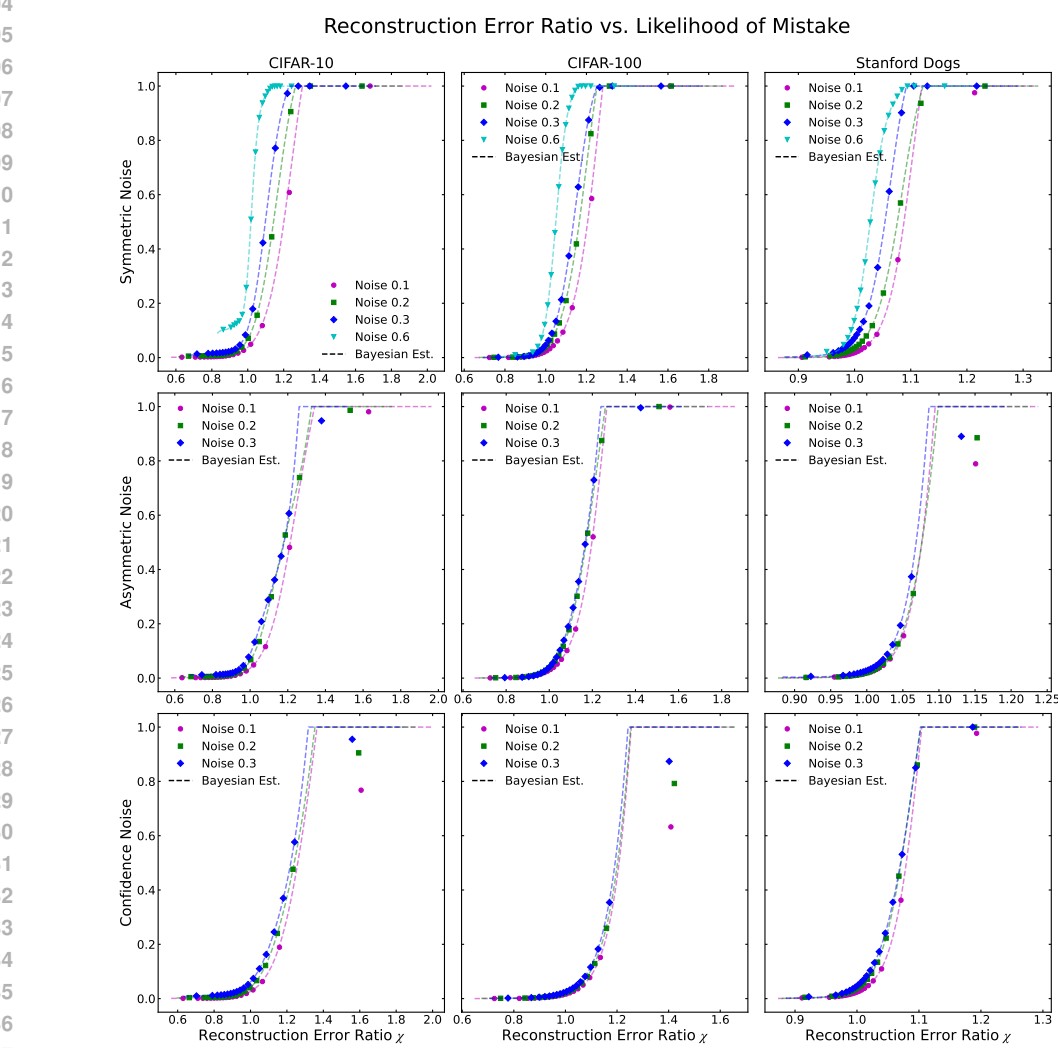

Figure 16: Likelihood of label mistakes as a function of reconstruction error ratio $\chi$ estimated from Eqs. (28) - (29), plotted against empirically estimated by binning mistake counts in the noisy labels. We use 20 evenly spaced bins.

The standard metric for evaluating binary classification tasks is the $F_1$-score:

$$F_1 := 2\frac{\text{precision} \times \text{recall}}{\text{precision} + \text{recall}} = \frac{2\text{TP}}{2\text{TP} + \text{FP} + \text{FN}}, \tag{31}$$

which is the harmonic mean of precision and recall.[6]

The simplicity of this formula hides the fact that the true positive, false positive, and false negative counts depend on the ground truth labels and predicted labels. To be more explicit, given a set of ground truth labels $Y^{\text{gt}} = \{y_j^{\text{gt}}\}_j$ and predicted labels $Y^{\text{pred}} = \{y_j^{\text{pred}}\}_j$, where $y_j = 1$ denotes a mistake and $y_j = 0$ denotes a clean sample,

---

[6]A common critique of the $F_1$-score is that it does not incorporate true negatives. The metric we define inherits this property as well. However, in practice, for the purposes of identification by mislabel, it serves as a relatively fair means of evaluation between different methods.

$$TP(Y^{\text{gt}}, Y^{\text{pred}}) = \sum_j y_j^{\text{pred}} \cdot y_j^{\text{gt}}, \tag{32a}$$

$$FP(Y^{\text{gt}}, Y^{\text{pred}}) = \sum_j y_j^{\text{pred}} \cdot (1 - y_j^{\text{gt}}), \tag{32b}$$

$$FN(Y^{\text{gt}}, Y^{\text{pred}}) = \sum_j (1 - y_j^{\text{pred}}) \cdot y_j^{\text{gt}}, \tag{32c}$$

And $F_1 \rightarrow F_1(Y^{\text{gt}}, Y^{\text{pred}})$.

Given confidence scores $W^{\text{pred}} = \{w_j\}_j$ for each prediction, we can extend Eqs. (32a)-(32c) as:

$$S_{TP}(Y^{\text{gt}}, Y^{\text{pred}}, W^{\text{pred}}) := \sum_j w_j \cdot y_j^{\text{pred}} \cdot y_j^{\text{gt}}, \tag{33a}$$

$$S_{FP}(Y^{\text{gt}}, Y^{\text{pred}}, W^{\text{pred}}) := \sum_j w_j \cdot y_j^{\text{pred}} \cdot (1 - y_j^{\text{gt}}), \tag{33b}$$

$$S_{FN}(Y^{\text{gt}}, Y^{\text{pred}}, W^{\text{pred}}) := \sum_j w_j \cdot (1 - y_j^{\text{pred}}) \cdot y_j^{\text{gt}}, \tag{33c}$$

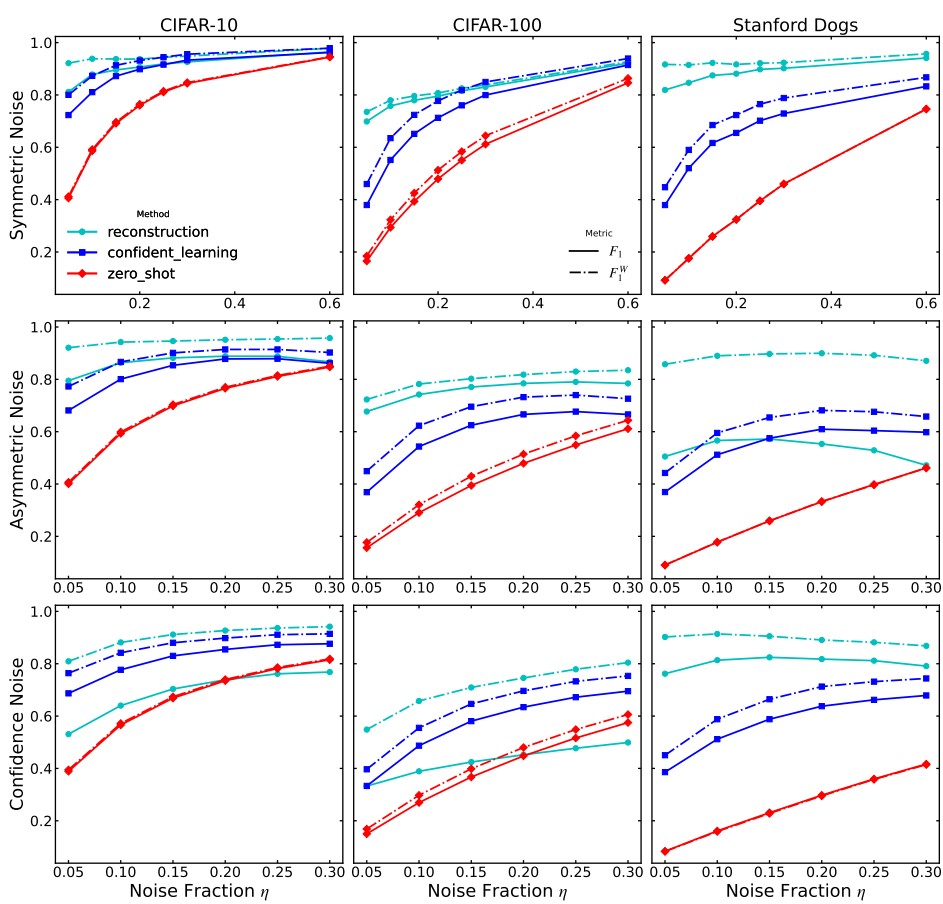

Figure 17: Comparison of standard $F_1$-score and confidence-weighted $F_1$-score for three exemplary datasets across three varieties of label noise.

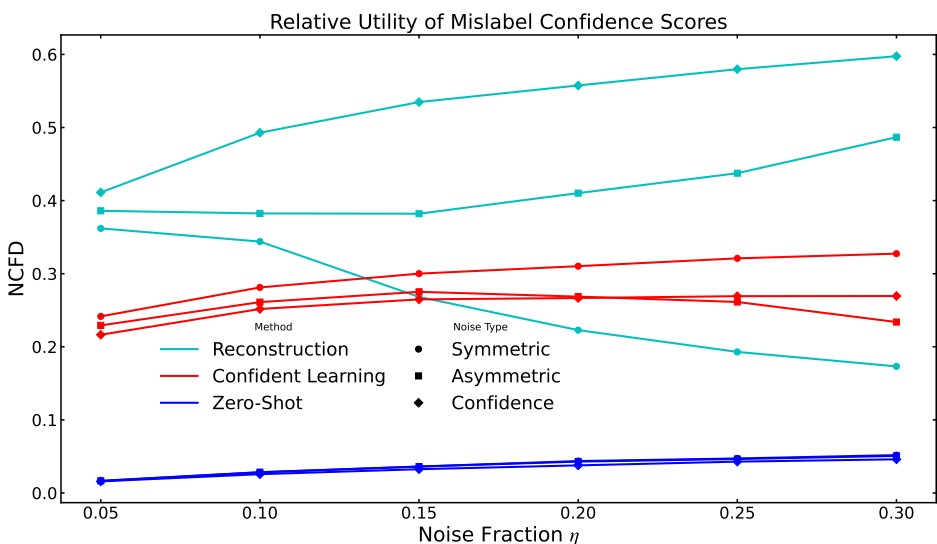

Figure 18: NCFD scores for mistakenness probabilities arising from reconstruction error ratios, confident learning, and zero-shot approaches for mislabel detection averaged over all datasets using CLIP ViT-B/32 features.

where $S_{TP}$, $S_{FP}$, and $S_{FN}$ are confidence-weighted sums, which place more emphasis on high-confidence predictions. Replacing our TP, FP, and FN counts with these confidence-weighted sums, we can define the *confidence-weighted $F_1$-score*:

$$F_1^W := \frac{2S_{TP}}{2S_{TP} + S_{FP} + S_{FN}}, \tag{34}$$

which reduces to the standard $F_1$-score in the limit $w_j = \text{const} \neq 0 \,\forall j$.

The relationship between $F_1$ and $F_1^W$ is illustrated in Fig. 17.

By comparing $F_1$ and $F_1^W$ for a fixed set of predictions, we can determine how much the confidence scores help in boosting performance. In particular, the *normalized confidence-weighted $F_1$ difference* (NCFD) is defined to be:

$$NCFD := \frac{F_1^W - F_1}{1 - F_1}, \tag{35}$$

where the numerator in Eq (35) is positive if confidence scores are beneficial, and negative if they detract from the baseline $F_1$-score. The denominator normalizes the gain in performance relative to baseline performance, allowing us to compare across different prediction methods and noise rates.

In practice, we compute the confidence scores from our probabilities as follows: given the probability threshold $p^*$ at which we begin to predict that a sample is mislabeled,

$$w_j = \begin{cases} \frac{p-p^*}{1-p^*}, & \text{if } p > p^* \\ \frac{p^*-p}{p^*}, & \text{otherwise,} \end{cases} \tag{36}$$

which symmetrizes across positive and negative predictions, even when the threshold is asymmetric.

We showcase the practical behavior of this quantity for three mislabel detection methods in Fig. 18, which illustrates that for asymmetric and confidence-based noise, as well as symmetric noise less than 20%, the probabilities generated by the RER framework are more helpful than those generated by Confident Learning.

*Proposition 1*: The confidence-weighted $F_1$-score defined in Eq. (34) is more sensitive to higher confidence predictions.

*Proof.* To prove this, let's suppose we have some initial set of ground truth labels and predictions $(Y^{\text{gt}}, Y^{\text{pred}}, W^{\text{pred}})_{[j]}$ for samples $1, \ldots, j$ resulting in an initial score, $F_{1,j}^W$. Consider the effect of adding a new triplet $(y_{j+1}^{true}, y_{j+1}^{pred}, w_{j+1}^{pred})$, and look at the resulting quantity $F_{1,j+1}^W$. For brevity, $S_{TP}$, $S_{FP}$, and $S_{FN}$ without an explicit index $j$ subscript will refer to the quantities involved in calculating $F_{1,j}^W$.

We have four cases to consider.

*Case I (False Negative)*: As $F_1$ and $F_1^W$ do not depend on false negative, we can safely ignore this case as trivial, and $F_{1,j+1}^W = F_{1,j}^W$.

*Case II (True Positive)*: In this case, $S_{TP} \to S_{TP} + w_{j+1}$, so

$$F_{1,j+1}^W = \frac{2(S_{TP} + w_{j+1})}{2(S_{TP} + w_{j+1}) + S_{FP} + S_{FN}}, \tag{37a}$$

$$= \frac{2(S_{TP} + w_{j+1})}{(2S_{TP} + S_{FP} + S_{FN})(1 + \frac{2w_{j+1}}{2S_{TP} + S_{FP} + S_{FN}})},$$

$$\approx \frac{2(S_{TP} + w_{j+1})}{2S_{TP} + S_{FP} + S_{FN}} \times \left(1 - \frac{2w_{j+1}}{2S_{TP} + S_{FP} + S_{FN}}\right), \tag{37b}$$

$$= \left(F_{1,j}^W + \frac{2w_{j+1}}{2S_{TP} + S_{FP} + S_{FN}}\right) \times \left(1 - \frac{2w_{j+1}}{2S_{TP} + S_{FP} + S_{FN}}\right) \tag{37c}$$

$$= F_{1,j}^W + (1 - F_{1,j}^W) \times \frac{2w_{j+1}}{2S_{TP} + S_{FP} + S_{FN}} + \mathcal{O}\left(\left(\frac{2w_{j+1}}{2S_{TP} + S_{FP} + S_{FN}}\right)^2\right), \tag{37d}$$

where in Eqs. (37b) and (37d) we use the fact that $\frac{2w_{j+1}}{2S_{TP} + S_{FP} + S_{FN}} \ll 1$, which will in practice be the case when the number of samples is of any substantial size.

Looking at the change in our confidence-weighted $F_1$-score,

$$\Delta_j F_1^W = F_{j+1}^W - F_1^W, \tag{38}$$

we have that for True Positive predictions,

$$\Delta_j F_1^W \approx (1 - F_{1,j}^W) \times \frac{2w_{j+1}}{2S_{TP} + S_{FP} + S_{FN}}, \tag{39}$$

which depends linearly on the prediction confidence.

*Case III and IV (False Positive/False Negative)*: the confidence-weighted $F_1$-score is symmetric with respect to false positive and false negative predictions, as adding either (with confidence $w_{i+1}$) will increase the denominator of Eq. (34) by $w_{i+1}$ and leave the numerator intact.

Employing the same approach from Case II, we find that:

$$F_{1,j+1}^W = \frac{2S_{TP}}{2S_{TP} + S_{FP} + S_{FN} + w_{j+1}}, \tag{40a}$$

$$= \frac{2S_{TP}}{(2S_{TP} + S_{FP} + S_{FN})(1 + \frac{w_{j+1}}{2S_{TP} + S_{FP} + S_{FN}})},$$

$$\approx \frac{2S_{TP}}{2S_{TP} + S_{FP} + S_{FN}} \times \left(1 - \frac{w_{j+1}}{2S_{TP} + S_{FP} + S_{FN}}\right), \tag{40b}$$

$$= F_{1,j}^W \times \left(1 - \frac{w_{j+1}}{2S_{TP} + S_{FP} + S_{FN}}\right), \tag{40c}$$

and plugging into (38),

$$\Delta_j F_1^W \approx -F_{1,j}^W \times \frac{w_{j+1}}{2S_{TP} + S_{FP} + S_{FN}}, \tag{41}$$

which is also proportional to $w_{j+1}$.

∎

