# OpenReview forum: "Class-wise Autoencoders Measure Classification Difficulty And Detect Label Mistakes"
_ICLR.cc/2025/Conference — Submitted to ICLR 2025_

### Official Review · Reviewer_gsPt · 2024-10-19

**Soundness:** 4
**Presentation:** 2
**Contribution:** 4
**Rating:** 8
**Confidence:** 3

**Summary:**

This paper introduces a novel approach for analyzing existing datasets by breaking down complex multi-class classification problems into class-wise reconstruction tasks. The proposed method shows strong performance in terms of interpretability, generality, and efficiency with some interesting observations.

**Strengths:**

1. The research topic is highly important and intriguing, especially in this data-centric era where data quality is more crucial than ever.
2. All the experiments and observations are fundamentally new and interesting.
3. The proposed method is simple but very intuitive. To borrow idea from ood detection is interesting.

**Weaknesses:**

#### Major problems
1. Although the observation and estimated rational function in Section 4.2 are informative, no further analysis of the dataset is provided. I'm curious whether the results in Figure 4 are based on a simple simulation or if there is some underlying rationale.
2. More detailed comparisons with existing method on quality assessment method should be included.
3. Some minor issues like grammar mistakes, spelling problems and wrong word order in the sentence make the reading a bit difficult.
4. Further discussions on applying the proposed method are needed. While classification is important, dataset quality assessment should encompass a wider range of research areas.

#### Minor issues
1. Missing word in line 85-86: "has long been challenge a in machine learning"
2. Grammar mistakes in line 46: "how well a new sample is reconstructed the shallow model"
3. Confusing expression in line 157-158: "which helps the very compact models learn the local and global structure of each class manifold"

**Questions:**

Some vague notations or conclusions: In lines 263-266, the authors claim that ViT-based features strongly predict classification dataset difficulty, whereas pretrained ResNet-style models are only weakly correlated. Further analysis would be interesting and important, as the choice of pretrained models is crucial for subsequent data assessment.

---

> ### Author Response · Authors · 2024-11-19
> **Clarifications on Practicality and Future Work**
>
> Thank you for your insightful review and for pushing us to make this work as strong as possible. We do our best to address your suggestions and feedback below.
>
> ### Weaknesses
> 1. As space in the paper was limited, we had to make difficult decisions about what to include in the main paper. However, we are eager to extend our analysis to more diverse scenarios. We also wish to clarify that each data point in Fig. 4 was generated by restricting the fitting of the autoencoders to a certain number of samples per class and running the entire reconstruction error ratio computation. We then used sklearn’s curve fitting functionality to estimate the coefficient in Eq. (5). Running this across many datasets, we found that the exponent in Eq. (5) seemed to be invariant, so we proceeded to pin that value. Fig. 7 in the original manuscript’s supplementary material (Fig. 8 in the revised manuscript) shows that the same functional form holds (to good approximation) for other reconstruction error ratios as well. What other analyses did you have in mind?
> 2. In our review of the literature, the other methods for classification difficulty that we identified required orders of magnitude more compute, or required tuning on a per-dataset basis (L83-100 of original paper). In the near future, we plan to directly and concretely compare optimized versions of the framework for mislabel detection and other dataset-level analysis tasks to existing state-of-the-art methods. What other quality assessment methods did you have in mind?
> 3. Thank you for finding these typos and helping us to improve readability! All such changes have been made and should be fixed in the revised manuscript.
> 4. Great question! We wholeheartedly agree that for reconstruction error ratios to be generally useful, they need to be applicable to a wider range of visual tasks. As we note in L527-531 of the original manuscript, future work will focus on extending this framework to object detection and segmentation tasks. We also note that our entire codebase will be made publicly available before the conference begins.
>
> ### Additional question regarding pre-trained models:
> The experiments supporting this paper involved three model families (ResNet, CLIP, and DINOv2). While there was strong alignment between CLIP and DINOv2 models, and both correlated strongly with reported error rates for state-of-the-art classification models on a per-dataset basis, we do not claim that vision transformers in general should strongly predict classification difficulty. By the same token, we do not claim that all non-transformer models are incapable of achieving strong performance in these tasks. Further analysis and exploration would be of great benefit and is left for future work.

---

> > ### Comment · Reviewer_gsPt · 2024-11-22
> >
> > Thanks for the further clarification, it has addressed my concerns and I will keep my score.

---

### Official Review · Reviewer_JvYp · 2024-10-29

**Soundness:** 2
**Presentation:** 2
**Contribution:** 2
**Rating:** 3
**Confidence:** 3

**Summary:**

This paper introduces a dataset classification difficulty analyzing approach using reconstruction error ratios (RERs) derived from autoencoders trained on individual classes. The study shows that RER-based difficulty metrics strongly correlate with state-of-the-art (SOTA) classification model error rates, which could be regarded as a probe for mislabeled detection tasks.

**Strengths:**

This paper starts an early attempt to leverage the prior knowledge of foundation models for classification difficulty analysis.

**Weaknesses:**

One major concern is that the writing and structure of this paper require careful improvement and reorganization. In the current version, numerous details are insufficiently introduced, which may impede understanding of the motivations and methods. For instance, in Fig. 1, the authors do not clearly define 'in-class' and 'out-of-class,' explain how $R^2$ is calculated, or discuss its implications. Additionally, the definitions of 'easiest,' 'median,' and 'hardest' class are unclear.

Another critical weakness is the limited contribution of this paper. It appears to mainly propose a new metric (Eq. 3) without designing a novel algorithm. As illustrated in Fig. 6 and Fig. 10, the performance of the proposed RER-based method is not particularly promising.

**Questions:**

Please refer to the Weakness.

---

> ### Author Response · Authors · 2024-11-19
>
> Thank you for the thoughtful review, and for the suggestions regarding readability, structure, and clarity in the paper. However, we find that the reviewer’s characterization of our method’s performance and contribution to the field does not accurately reflect the actual substance of our work. As per reviewer gsPt, “The proposed method shows strong performance in terms of interpretability, generality, and efficiency”, and per reviewer hcLW, “The paper tackles an important problem in data-centric machine learning, which is to analyse the characteristics of datasets”. We address the reviewer’s questions below.
>
> In response to your suggestions, we have added clarifications and explanations for all aspects of Fig. 1 that you have brought to light. Where else do you think the paper could benefit from clearer and more concrete explanations?
>
> As for your second point, while we do not introduce a new model, so to speak, our paper attempts to introduce a new framework for understanding and analyzing visual datasets, which according to reviewer LCSk are “novel”, “straightforward”, and “more effective than previous ones”. Per the [ICLR 2025 guidelines](https://iclr.cc/Conferences/2025/ReviewerGuide), contributions need to “convincingly demonstrate new, relevant, impactful knowledge,” which can be theoretical and/or empirical. Concretely, here are three ICLR 2024 award winning or honorable mention papers that similarly contribute via a combination of theoretical and empirical results without training a new state-of-the-art model:
> - [Generalization in diffusion models arises from geometry-adaptive harmonic representation](https://openreview.net/forum?id=ANvmVS2Yr0)
> - [Beyond Weisfeiler-Lehman: A Quantitative Framework for GNN Expressiveness](https://openreview.net/forum?id=HSKaGOi7Ar)
> - [Towards a statistical theory of data selection under weak supervision](https://openreview.net/forum?id=HhfcNgQn6p)
>
> We disagree with the characterization of the RER framework’s performance shown in Fig. 6 and Fig. 10 of the original paper (6 and 13 in the revised paper) as lacking promise. In Fig. 6, we show that with a simple ansatz, RERs already enable SOTA or near-SOTA mislabel detection under symmetric and asymmetric label noise. Fig. 13 in the revised manuscript illustrates the fact that even when specific thresholds are taken out of the equation, RER-based mislabel detection performs on par with or better than established methods like confident learning for symmetric and asymmetric noise. The method’s performance on confidence-based noise can very likely be improved via architectural changes, which we leave for future work.
>
> We believe that initiating the principled analysis of datasets via reconstruction error ratios and empirically demonstrating the utility of this approach for classification difficulty and mislabel detection represents a significant contribution to the field.

---

### Official Review · Reviewer_jSA7 · 2024-10-30

**Soundness:** 2
**Presentation:** 3
**Contribution:** 3
**Rating:** 6
**Confidence:** 3

**Summary:**

The authors study dataset classification difficulty via  , find strong correlation with classification error rate of SOTA models, while apply the sample-level classification difficulty as a label mistakenness score to detect noisy labels, finally via relatively sufficent empirical analysis comfirms their mitivation.

**Strengths:**

1. For analysing dataset classification difficulty, the authors first define a difficulty measure via RERs from AE reconstruction errors and build so-called new analysis theory and  claim the RER is model-agnostic.
2.  Based on the above measure, it can be decomposed to two parts: (1) finite sample size and (2) Bayes error and decision-boundary complexity.
3. Via a systemic experimental study, the authors find some insightful results, such as strong correlation with classification error rate and mislabel detection.

**Weaknesses:**

1. the authors claim their theory is model-agnositc, however, actually, dependent on AE model since datasets with different difficulties will more likely result in the same separability, which comes from strong modeling ability of AEs, the latter makes many difficult datasets prone to linear separable, yielding possible unidentifiability!
2. Based on the above 1, datasets may form some equivalance groups in difficulty measure, for which needing the authors to analysis.
3. From its difinition, RERs seems NOT robust, the authors do NOT clarify it!

**Questions:**

1. the setting discussed is NOT that practical, the authors assume classes in training are balanced, however, often assuming long-tailed character. while from their discussion, it seems the inputs are clean though labels are noisy, which is too ideal, noisy labels are exactly more likely from unclean inputs such as images.
2. Insufficient analyses for RERs, such as scaling invariance and robustness to noise.
3. Do those representations from different foundation models such as CLIP and DINOv2 you use here have different "difficulty" scores or correlation?  Or are their "difficulties" consistent? for example, samples' or classes's difficulties are order-preserving?
4. At lines 149-150, "decompose the dataset by class and use shallow autoencoders to learn robust representations of these class manifolds" appears NOT to using classification, however, essentially, using class-wise autoencoders to learn representaions corresponding to individual classes has infused class information!  whether could you try to re-determine the difficulty by artificially merging two-classes to one meta-class?
5. when training the c-th encoder using mislabeled x^c (in fact, x^c has ground-truth class c'), the encoder will be brittle to the noisy sample!, but the mislabeled x^c  may still have small reconstructed error, naturally yielding misleading error!

---

> ### Author Response · Authors · 2024-11-19
> **Remarks on Consistency, Robustness, and Generality**
>
> Thank you for the comprehensive review. We respond to each of your questions and comments below:
>
> ### Weakness
> 1: While the reconstruction error ratio framework does require autoencoders, our framework is model-agnostic in the following senses:
> - None of the pre or post-processing of samples needs to be changed from dataset to dataset (L57-59 in the original paper)
> - No custom architectural decisions need to be made for a given dataset
> - None of the analysis depends on the autoencoder or the feature backbone, and
> - Results are remarkably consistent across the CLIP and DINOv2 feature backbones (Fig. 9 in the original paper and Fig. 11 and Table 3 in the revised paper), which were trained in very different fashions. The autoencoders we use are shallow, yet we see strong correlation between the estimated difficulty and SOTA classification performance across a wide range of datasets.
> 2. Given our response to 1, we believe that your second question may no longer be a concern. If it is, can you clarify the question?
> 3. From our experiments, we believe RERs to be quite robust. We observed consistency and robustness across random initialization (as described in L397-398 and L453-454 in the original paper), error settings, feature backbones (Fig. 9 in the original paper, and Fig. 11 and Table 3 in the revised paper), and autoencoder hyperparameters (L767-790 in the original paper, and supplemented by Fig. 7 in the revised paper).
>
> ### Questions
> 1. We care deeply about applying this technique in practical settings, and plan to investigate a wide range of edge cases in future work (L527-531 in the original manuscript). We also want to add clarity to two points brought up by the reviewer:
>  - The only place where we assume balanced datasets is in Fig. 4 and the discussion around extrapolation to the infinite size limit. All other analysis include datasets with varying degrees of class imbalance. The MIT Indoor Scenes dataset, for example, has one class with 101 instances and another with 734 instances, and SUN397 has classes with 100 instances and 2361 instances.
>  - Foundation representation models like CLIP and DINOv2 can handle some degree of noise in the images themselves. We expect that the RER framework would inherit robustness to image noise from these feature extraction models.
> 2. Figures 4 and 7 in the original paper (4 and 8 in the revised paper) show the scaling of estimated dataset difficulty based on the number of samples per class used to fit each autoencoder, and Fig. 12 in the original paper (15 in the revised paper) shows the scaling of RER-based mislabel detection on the fraction of samples used. Additionally, per your suggestion we have added a hyperparameter sweep analysis (Fig. 7, L846-864 in revised manuscript) to demonstrate the framework’s robustness to hyperparameter settings. Beyond these highlighted scaling analyses, what did you have in mind? What robustness analyses would you recommend performing?
> 3. Great question! In response to this question, we have conducted further analysis on the alignment/consistency in ordering of RERs across various features, both on the sample and class levels. These analyses are summarized in Fig. 10 and Table 3 in the revised manuscript. In summary, we find that CLIP and DINOv2 backbones are moderately correlated when considering all samples/classes (Spearman Rank correlation coefficient), and when weighting the highest-scoring samples/classes more strongly (nDCG) they are highly correlated.
> 4. You are absolutely correct that by fitting the autoencoders on class-specific features, we are infusing class information into the process. This is by design. One could certainly perform experiments with artificially merged classes or classes at different hierarchical/semantic levels. We leave this for future work.
> 5. In theory, including mislabeled samples in the training data for an autoencoder could lead to artificially low reconstruction error for that specific sample. Our RER framework ensures a high degree of robustness by keeping the autoencoders shallow, regularizing with UMAP loss (which prioritizes overall graph layout considerations), and performing early stopping. In practice we do not see this issue arise — across 19 distinct datasets we observe the ability to estimate dataset classification difficulty, and strong performance on mislabel detection tasks, as documented in the paper.

---

> > ### Comment · Reviewer_jSA7 · 2024-11-26
> > **Class-wise Autoencoders Measure Classification Difficulty And Detect Label Mistakes**
> >
> > Thanks for your responses to my comments and read them relatively carefully, I decide to still keep the original score!

---

### Official Review · Reviewer_hcLW · 2024-10-30

**Soundness:** 2
**Presentation:** 3
**Contribution:** 1
**Rating:** 3
**Confidence:** 3

**Summary:**

The paper propose a novel method based on the reconstruction loss of class-wise samples to analyse several datasets used in classification. In particular, the proposed method employs features extracted from some pre-trained models to train class-wise auto-encoders and use the normalised reconstruction loss, either at sample level or dataset level, to analyse different characteristics of datasets, such as classification difficulty or label noise detection. The paper also provides some post-hoc analysis that matches several observations from a wide-range of visual classification datasets.

**Strengths:**

The paper tackles an important problem in data-centric machine learning, which is to analyse the characteristics of datasets. The major strength is to provide an extensive empirical results and several observations on many visual datasets used in classification.

**Weaknesses:**

1. The paper is purely an empirical study based on many observations from a certain number of datasets. However, the claim is to provide a formal theory to analyse datasets, to me, is exagerating. This may be because there are different understanding of theory. To my understanding, theory is developed from certain axioms, then theorems built uppon those axioms coupled with further assumptions are proved. However, the ones presented in the paper are only empirical observations without any formal proofs. Some of the analysis, to me, are just curve-fitting based on what have been observed (e.g., Eq. (5)). To this end, I disagree with the claim about a new theory for dataset analysis.
2. The authors make many strong confirmations based on empirical observations on a narrow setting without any formal proof. For example, at line 262, the conclusion that the "specific features used are irrelevant". This may be true in this setting because the two feature extractors (e.g., CLIP and DINO) are pre-trained on datasets that already include or have similar images in the 18 datasets considered. If a bad feature extract is used, the empirical result may be different. In addition, if a dataset that is outside of the training data of those feature extractors, the results may also be different. For example, one can evaluate on datasets in medical, biology or space exploration.
3. In section 4.3 presenting a way to estimate symmetric label noise from the reconstruction error ratio, the proof requires further clarification, especially when simplifying the second term in Eq. (16). To make it clear, the second term of Eq. (16) is restated here:
$$
\text{second term of (16)} = \eta \mathbb{E}_{X} \left[ \frac{\Delta\_{\tilde{c}} (\mathbf{x}^{c^{\prime}}) }{ \Delta\_{rand} (\mathbf{x}^{c^{\prime}}) } \right],
$$
and the one defined in Eq. (6) is:
$$
\Delta\_{rand} (\mathbf{x}^{\tilde{c}}) = \mathbb{E}\_{c^{\prime} \in \mathcal{C} \textbackslash {c}} \left[ \Delta\_{c^{\prime}} (\mathbf{x}^{\tilde{c}})  \right].
$$
It is hard to understand how to use the result in Eq. (6) to apply to the numerator of the second term in Eq. (16) because:
 - the two expectations are different. The one in (16) is w.r.t. $\mathbf{x}$, while the one in (6) is w.r.t. auto-encoder or class $c^{\prime}$, and
 - the expectation in (16) is applied to a ratio of two functions of $\mathbf{x}$, and note that: $\mathbb{E}[f(x) / g(x)] = \mathbb{E}[f(x)] / \mathbb{E}[g(x)]$ does not generally hold.

**Questions:**

My main concerns have been stated in the weaknesses. Please address and provide further clarifications.

---

> ### Author Response · Authors · 2024-11-19
> **Additional Analyses for Generalization and Explanation of Proof**
>
> Thank you for kindly acknowledging the importance of and need for dataset-level analysis in machine learning, and for giving us the opportunity to refine our terminology and arguments. In the process of writing the manuscript, we went back and forth several times on the notion of theory vs framework. Your perspective confirms that framework is a better description of our contribution, which is mathematically motivated but backed up entirely by empirical evidence. We have revised the paper with this distinction in mind. We address your questions/comments below:
>
> ### Questions
> 1. As mentioned above, per your suggestion we have revised our terminology from theory to framework. Thank you for the suggestion.
> 2. Thank you for helping us to better contextualize our findings. It is indeed very likely the case that foundation models like CLIP and DINOv2 were trained in part on datasets like those in our study. We agree that it would be unreasonable to expect these models to apply equally well to arbitrary visual classification tasks, and that bold claims of the sort that you reference require reframing. In the revised manuscript, we change “irrelevant” to “immaterial”. Thank you as well for the suggestion to test this framework on datasets one would expect to be out of domain for models like CLIP and DINOv2. In response to your suggestion, we have employed the reconstruction error ratios framework on 10 medical datasets from MedMNISTv2 (only omitting the 2 that are pure binary classification tasks). We note that a direct comparison between these medical datasets and the 18 datasets in our main study is not straightforward because Fig. 3 uses state-of-the-art classification accuracy, and for MedMNIST datasets, we could not find reliable resources for these values. Nevertheless, we observe the same trend when plotting an analogous version of Fig. 3 for MedMNISTv2 using the best classification accuracy values reported in the [MedMNIST v2 technical report](https://arxiv.org/pdf/2110.14795) (Fig. 12 in the revised manuscript, L1164-1188).
> 3. Thank you for giving us the opportunity to make our reasoning more explicit. Your question was regarding the jump from Eq. (16) to Eq. (17), particularly focused on second term, which we denote $T_2$:
>
> $\displaystyle T_2 = \eta  \, 𝔼_{X}\big[ \frac{\Delta_{\tilde{c}}(\boldsymbol{x}^{c'})}{\Delta_{rand}(\boldsymbol{x}^{c'})}\big],$ which obtains when sample $\boldsymbol{x}$ is mislabeled such that the true class for the sample is $c' \neq c$.
>
> The only dependence on $\tilde{c}$ is now in the noisily fitted autoencoder in the numerator, and we already know that $c \neq c'$ so we can approximately expand this as an average over all noisily fitted autoencoders for classes other than sample $\boldsymbol{x}^{c'}$:
>
> \begin{align*}
>     T_2 = \eta  \, 𝔼_{X}\Big[  \frac{𝔼_{c'' \in \mathcal{C} \backslash \{ c'\}}\big[\Delta_{\tilde{c}''}(\boldsymbol{x}^{c'})\big]}{\Delta_{rand}(\boldsymbol{x}^{c'})}\Big].
> \end{align*}
>
> We can recognize the numerator as $\Delta_{rand}(\boldsymbol{x}^{\tilde{c}})$, which was defined as $ \Delta_{rand}(\boldsymbol{x}^{\tilde{c}}) = 𝔼_{c' \in \mathcal{C} \backslash \{ c\}}[\Delta_{\tilde{c'}}(\boldsymbol{x}^{\tilde{c}})]$, giving us the expression: $T_2 = \eta  \, 𝔼_{X} \Big[ \frac{\displaystyle \Delta_{rand}(\boldsymbol{x}^{\tilde{c}})}{\displaystyle \Delta_{rand}(\boldsymbol{x}^{\tilde{c}})}\Big]$, which resolves to $T_2 = \eta$.
>
> Intuitively, the resulting approximate expression (19) in the paper: $ \chi_0 \approx (1 - \eta)  \chi_{rand} + \eta$ can be understood as follows:
>
> - $\chi_0$ represents how much better the supposed ground truth label fits a sample than a random class in the dataset
> - $\chi_{rand}$ represents how much better the *best* fitting class label is at representing a sample than a random class
> - $\eta$ represents the noise, from both mistakes and ambiguity, in the dataset
>
> Expression (19) in the paper gives rise to the interpretation that we can approximate the noise in the dataset by looking at the differences between how much better than random the best class is at representing a sample than the supposed ground truth label class.

---

> > ### Comment · Reviewer_hcLW · 2024-11-25
> > **Inadequate approximation**
> >
> > Thank you, the authors, for the clarification. However, the approximation to explain Eq. (16) does not make sense. Simply approximating a (equivalent) random variable by its expected value cannot be justified. Hence, the result claimed from Eq. (16) to Eq. (17) is invalid as mentioned in the initial comment.
> >
> > Given that the paper mostly presented empirical results and some inadequate statements, I keep my rating as is.

---

> ### Author Response · Authors · 2024-11-25
> **Further Clarification**
>
> We thank the reviewer for taking the time to read and respond. We wish to further clarify a few points:
>
> 1. In our initial response, we addressed your questions about the generality of our method by applying RER-based difficulty estimation to 10 out-of-domain medical datasets, with great success.
> 2. As far as the efficacy of our method is concerned, the proof in Appendix B.2 only influences the selection of a threshold in our self-reportedly rough ansatz for mislabel detection. It does not factor into difficulty estimation at all, nor does it influence the AUROC of RER-based mislabel detection.
> 3. We believe that our substitution of variables in the proof is justified, under the assumption that these variables are normal-distributed, and given that we are looking at performance in the mean.
>
> Thanks again for your consideration.

---

### Official Review · Reviewer_LCSk · 2024-11-04

**Soundness:** 3
**Presentation:** 3
**Contribution:** 3
**Rating:** 8
**Confidence:** 4

**Summary:**

This work presents a method for noisy label detection, which also serves as a way to assess dataset difficulty. Using a large visual foundational model (non-feature-based, so not convolutional), the method extracts representations for each datapoint. Next, simple autoencoders with UMAP regularization are trained for each class. The ratio between reconstruction errors for the target class and the next-best autoencoder class is then used to estimate label noisiness and misclassification probabilities. From my understanding, this ratio between reconstruction errors acts as a proxy for margin. Furthermore, the authors provide numerical results to help predict the inherent difficulty of a class by interpolating reconstruction ratios at an infinite number of datapoints.

**Strengths:**

The paper proposes a simple yet (according to the authors) novel method to approximate dataset difficulty with minimal training required on basic autoencoder models. Furthermore, their methods for detecting misclassification are straightforward and reportedly more effective than previous ones.

**Weaknesses:**

Much of the issues I have with the paper lie in the presentation of the work, which may have led me to misunderstand certain concepts. For example, when explaining the different types of noise (i.e., symmetric, confidence-based, etc.), I still don’t feel I have a solid grasp of these concepts after reading the passage. Perhaps the authors could clarify further, either with mathematical notation or visuals, to help readers gain a better understanding. Another example is the appearance of the tuning parameters, such as $\gamma$, which are often introduced but not discussed in depth.

With regard to the graphs, while I understand and acknowledge the difficulty of fitting so much information into limited space, I think the visuals could be improved. Figures 1 and 5–6 are particularly cluttered due to the choices of pointers and line styles for different noise types and methods, making them difficult to interpret. Meanwhile, I think the space occupied by Figure 2 could be better used to provide additional information for the reader.

Finally, regarding the experimental setup, while I understand that details are provided in the supplementary material, I would appreciate further discussion in the main body on certain neural architecture specifics. Additionally, I find it odd that certain larger datasets, such as ImageNet, are excluded from the analysis—though I may have missed this explanation.

**Questions:**

1- Doesn’t Equation 9 supposedly provide a form for retrieving the threshold? Are these hyperparameters expected to work universally across any dataset? If so, what is the purpose of Section C.1?

2- Could you please provide a mathematical or visual explanation of the different noise types?

3- Are there specifics regarding whether a particular datapoint has appeared in the training set of the foundational model used? Wouldn't this have certain theoretical and practical implications if the datapoint or even the class hasn’t appeared in the source dataset of the foundational model?

4- Is the notion of the ratios of reconstruction errors effectively a reinterpretation of classification margin? Do they have a direct relationship, such as an upper or lower bound?

5- To find the extrapolated limits for class difficulty, don’t we need to train with a variety of different sample sizes, and wouldn’t this increase the algorithm’s cost significantly? If I understand correctly, this increased cost doesn’t apply to label noise detection, correct?

6- In Figure 1, this is perhaps a minor detail, but why are the higher values (brighter points) not located on the edges of the distribution in the 2D representation? Shouldn't they be more outlier-like compared to other datapoints?

---

> ### Author Response · Authors · 2024-11-19
>
> Thank you for the thorough and thoughtful review! We address your comments below:
>
> ### Weaknesses
> 1. We appreciate you calling attention to the absence of ImageNet in the paper. While running experiments for this project, we were not able to obtain the original ImageNet dataset from imagenet.org. We have retrieved the dataset from colleagues based on your comment, and have added ImageNet into Fig. 3 and the accompanying dataset classification difficulty analysis, as well as Table 3 in the supplementary material of the revised paper. We find that ImageNet fits well with the picture we have painted. With ImageNet in Fig. 3, we can also see the trend that datasets with many classes (ImageNet, SUN397, Places205) tend to have slightly lower top-1 accuracy than their estimated classification difficulty would suggest, which we attribute to a focus in the computer vision community on top-5 accuracy in such scenarios.
> 2. Thank you for your suggestion to improve the visual presentation of our figures. Based on your feedback, we’ve slimmed down Fig. 2 to leave more room for exposition.
>
> ### Questions
> 1. Great question! Eq. (9) (lines 465-467 in original paper) is a heuristic functional form that we found to be pretty good at giving us a threshold — given our computed chi and our estimated eta, this equation gives a threshold that is decent. We chose this form because it is simple and satisfies a few properties of interest, as detailed in the supplementary material. However, this ansatz is far from perfect. The construction of this simple ansatz is meant to demonstrate that reconstruction error ratios can be applied to mislabel detection. The fact that with this simple ansatz we still achieve SOTA performance on mislabel detection under symmetric and asymmetric noise is a further indication that the approach is promising. Nevertheless, there is much room for improvement in this particular regard.
> 2. Thanks for the suggestion! In the revised version, we have added explanations for all four varieties of label noise to Sec. 5 (L473-486).
> 3. It is very difficult to say for certain whether or not a specific dataset or its samples appeared in the training data for foundation models like CLIP and DINOv2. This information does not seem to be made public. This could certainly impact theoretical guarantees. However, our expectation was that by showing that the RER framework exhibits similar behavior for multiple model architectures with differing training data and processes, we have demonstrated practical viability - this is why we applied the approach to so many datasets.
> 4. Great question! RERs capture many varied signals including, to your point, an important signal — classification margins. RERs additionally enable rich analyses, such as Eq. (8) in our paper, which gives an estimate for the noise rate in a dataset based on not just the null hypothesis (supposed ground truth label) and best alternative hypothesis (lowest reconstruction error), but also their relations to how a randomly chosen reconstructor would fare when reconstructing the given sample’s features. We plan to formalize the relationship between RERs and classification margins in future work. To this end, we have included a note about investigating this connection in the revised manuscript (L1071-1073).
> 5. The exercise in extrapolating the classification difficulty score to the infinite size limit is achieved by picking multiple sizes and, for each size, restricting the fitting of autoencoders for each class to a subset of that size. There is no simulation here, this involves running the RER computations once per data point in Fig. 4, which we have performed as a one-time experiment. The experiment demonstrates that 100 samples per class are generally sufficient for approximately assessing the classification difficulty of a dataset. If one is interested in estimating the infinite-size behavior for a particular dataset, they can run this experiment on said dataset. However, this analysis is not necessary for estimating the classification difficulty of the dataset as a whole or for detecting label mistakes. We have added a clarifying note to L908-9190 in the supplementary material — thank you for this question.
> 6. Awesome question! In the previous version of Fig. 2, the higher values tend to occur either A) near the boundaries between classes or B) for samples that are intrinsically mislabeled in the dataset. To elaborate, while the clusters aren’t colored by class label, if you squint, you can make out 10 distinct clusters in the figure. Many of the most “difficult” samples are on the edge of one cluster or in between two clusters. Additionally, even popular visual datasets like CIFAR10 are known to have label mistakes, which are likely the few bright points that show up at the center of clusters in the figure. As noted above, we have removed this component from the figure to leave more room for explanations.

---

> > ### Comment · Reviewer_LCSk · 2024-11-27
> >
> > I would like to thank the authors for their careful consideration of my remarks and the inclusion of new datasets and results in the paper. Given the changes and after considering the points raised by other reviewers, I have decided to raise my score.

---

### Author Response · Authors · 2024-11-19
**General Remarks**

We thank the editor, five reviewers, and area chair for reviewing our paper. The reviewers commented that:
- “The research topic is highly important and intriguing, especially in this data-centric era where data quality is more crucial than ever”, and “All the experiments and observations are fundamentally new and interesting” (gsPt)
- “The paper proposes a simple yet (according to the authors) novel method to approximate dataset difficulty with minimal training required on basic autoencoder models. Furthermore, their methods for detecting misclassification are straightforward and reportedly more effective than previous ones.” (LCSk)
- “The paper tackles an important problem in data-centric machine learning, which is to analyse the characteristics of datasets. The major strength is to provide an extensive empirical results and several observations on many visual datasets used in classification.” (hcLW)
- “This paper starts an early attempt to leverage the prior knowledge of foundation models for classification difficulty analysis.” (JvYp)
- “Via a systemic experimental study, the authors find some insightful results, such as strong correlation with classification error rate and mislabel detection.” (jSA7)

We thank the reviewers for their positive comments.

The paper’s stated contributions include:
1. A formal framework for applying Reconstruction Error Ratios for dataset analysis
2. Empirical validation of RERs as a measure of the difficulty of classification
3. A method for decomposing classification difficulty into distinct components representing
 finite-size contributions and Bayes error and decision-boundary contributions.
4. Demonstration of RERs’ efficacy in mislabel detection tasks.

The reviewers’ primary concerns were regarding the presentation and generality of our results. In response to the former, we have clarified the contents of Figs 1 and 2, and have added definitions of all four varieties of noise considered in Sec. 5. In the original paper, we sought to demonstrate generality via systematic study across 18 datasets (Fig. 3), analyzing the correlation in estimated difficulty across feature backbones (Fig. 9 in original paper), and showing how performance on mislabel detection tasks scaled as a function of the fraction of data used to fit class-wise autoencoders (Fig. 12 in original paper). In response to the reviewers’ comments, we have further backed up our findings as follows:
1. Adding ImageNet into our analyses (Fig. 3 and Table 3 in revised paper), in response to reviewer LCSk’s comments, showing that the RER framework applies to datasets with 1000+ classes.
2. Applying the RER framework to and adding new results for 10 medical datasets (Fig. 12 in revised paper) in response to reviewer hcLW’s suggestion, and observing the same trends in an entirely different domain, showing that our framework applies beyond the assumed training data distribution of the feature backbones utilized.
3. Extending our comparison of feature backbones to include analyses of correlations on the sample and class levels (Fig. 11 and Table 3 in the revised paper) in response to reviewer jSA7’s suggestions, further highlighting the generalizability of our results.
4. Adding hyperparameter sweep analyses (Fig. 7 in revised paper) in response to reviewer jSA7’s questions around robustness, demonstrating that the RER framework is robust to a wide range of hyperparameter settings.

All changes made to the manuscript have been highlighted in blue for ease of reference, and this coloring will be removed in the final version. We greatly appreciate the suggestions of all five reviewers and the area chair and believe that our revised paper presents an even more compelling contribution to the field.

---

### Meta-Review · Area_Chair_dRPE · 2024-12-10

**Metareview:**

This paper proposes a new metric to measure classification difficulty based on class-wise autoencoders, called reconstruction error ratios (RERs). Empirical investigation shows that RER strongly correlates with error rate for state-of-the-art classification models. The method can also be used to detect noisy labels in a sample-wise fashion.

The reviewers agree that the problem of dataset difficulty is important, and the proposed approach is simple and easy to understand. Experiments show correlations with SOTA classifier performance, and can be used to detect mislabeled samples.

On the other hand, the main concerns of reviewers were:
1. The term "theory" is used to explain the contribution but the paper is mostly empirical. (The authors modified the term in the paper during the rebuttal phase.)
2. Strong claims based on only empirical study/observations. (the authors cite the ICLR 2025 guidelines and explain how contributions can be empirical)
3. The paper does not propose a "framework", just an empirical observation or a metric. (This was a concern that came up during the AC-reviewer discussion, so te authors didn't have a chance to respond to this.)
4. Only a new metric but no new algorithm (The rebuttal argues that this is acceptable according to the ICLR 2025 guidelines, and also mention a few award-winning papers from ICLR 2024 that have a similar taste)
5. Experiments with medical, biology, or space exploration is a plus (The authors added 10 medical datasets)
6. A concern about the assumptions in the proof in the Appendix (The authors say that the proof is justified, under the assumption that these variables are normal-distributed, and given that we are looking at performance in the mean. It only influences the selection of a threshold for mislabel detection.)

Based on the rebuttal and discussions, most of these concerns seemed to be addressed, as explained inside the parenthesis above. My further comments for the 2nd and 3rd point:

2: I agree with the authors that contributions can be empirical in ICLR. Nonetheless, I suggest that the authors adjust the tone to clearly distinguish between empirical observations and more general claims, to avoid overstating the scope of their findings.

3: this came up during the AC-reviewer discussion. I agree with the reviewer that brought this up, but this is somewhat subjective, and the authors may choose to retain the term.

Furthermore, we noticed that the current version of the paper lacks discussions about many similar work especially in the area of Bayes error estimation and instance difficulty estimation. For example, "Evaluating State-of-the-Art Classification Models Against Bayes Optimality" (NeurIPS 2021) is a very similar paper with similar results. This paper already shows how the estimated Bayes error and SOTA classification error correlate. Although the details of the proposed methods are different, the conceptual novelty may be limited since the approach is similar at a high level (both rely on generative modelling) and similar insights regarding SOTA correlation.

Based on the reviews, rebuttal, post-rebuttal discussions, and the meta review we have written so far, we would like to recommend rejection. We encourage the authors to discuss the novelty aspect (by comparing to the paper mentioned in the previous paragraph) in detail to clarify the novelty in future versions of the paper.

**Additional Comments On Reviewer Discussion:**

I summarized the main points raised by the reviewers above. There were 2 reviewers with rating 3, and one of them did not reply to the rebuttal.

---

### Decision · Program_Chairs · 2025-01-22

Reject